# GEODIFF: A GEOMETRIC DIFFUSION MODEL FOR MOLECULAR CONFORMATION GENERATION

**Minkai Xu**[1,2]**, Lantao Yu**[3]**, Yang Song**[3]**, Chence Shi**[1,2]**, Stefano Ermon**[3*]**, Jian Tang**[1,4,5*]

[1]Mila - Québec AI Institute, Canada [2]Université de Montréal, Canada
[3]Stanford University, USA [4]HEC Montréal, Canada [5]CIFAR AI Research Chair
`{minkai.xu,chence.shi}@umontreal.ca`
`{lantaoyu,yangsong,ermon}@cs.stanford.edu`
`jian.tang@hec.ca`

## ABSTRACT

Predicting molecular conformations from molecular graphs is a fundamental problem in cheminformatics and drug discovery. Recently, significant progress has been achieved with machine learning approaches, especially with deep generative models. Inspired by the diffusion process in classical non-equilibrium thermodynamics where heated particles will diffuse from original states to a noise distribution, in this paper, we propose a novel generative model named GEODIFF for molecular conformation prediction. GEODIFF treats each atom as a particle and learns to directly reverse the diffusion process (*i.e.*, transforming from a noise distribution to stable conformations) as a Markov chain. Modeling such a generation process is however very challenging as the likelihood of conformations should be roto-translational invariant. We theoretically show that Markov chains evolving with *equivariant* Markov kernels can induce an *invariant* distribution by design, and further propose building blocks for the Markov kernels to preserve the desirable equivariance property. The whole framework can be efficiently trained in an end-to-end fashion by optimizing a weighted variational lower bound to the (conditional) likelihood. Experiments on multiple benchmarks show that GEODIFF is superior or comparable to existing state-of-the-art approaches, especially on large molecules.[1]

## 1 INTRODUCTION

Graph representation learning has achieved huge success for molecule modeling in various tasks ranging from property prediction (Gilmer et al., 2017; Duvenaud et al., 2015) to molecule generation (Jin et al., 2018; Shi et al., 2020), where typically a molecule is represented as an atom-bond graph. Despite its effectiveness in various applications, a more intrinsic and informative representation for molecules is the 3D *geometry*, also known as *conformation*, where atoms are represented as their Cartesian coordinates. The 3D structures determine the biological and physical properties of molecules and hence play a key role in many applications such as computational drug and material design (Thomas et al., 2018; Gebauer et al., 2021; Jing et al., 2021; Batzner et al., 2021). Unfortunately, how to predict stable molecular conformation remains a challenging problem. Traditional methods based on molecular dynamics (MD) or Markov chain Monte Carlo (MCMC) are very computationally expensive, especially for large molecules (Hawkins, 2017).

Recently, significant progress has been made with machine learning approaches, especially with deep generative models. For example, Simm & Hernandez-Lobato (2020); Xu et al. (2021b) studied predicting atomic distances with variational autoencoders (VAEs) (Kingma & Welling, 2013) and flow-based models (Dinh et al., 2017) respectively. Shi et al. (2021) proposed to use denoising score matching (Song & Ermon, 2019; 2020) to estimate the gradient fields over atomic distances, through which the gradient fields over atomic coordinates can be calculated. Ganea et al. (2021) studied generating conformations by predicting both bond lengths and angles. As molecular conformations are roto-translational invariant, these approaches circumvent directly modeling atomic coordinates by leveraging intermediate geometric variables such as atomic distances, bond and torsion angles, which

---

[1]Code is available at `https://github.com/MinkaiXu/GeoDiff`.

are roto-translational invariant. As a result, they are able to achieve very compelling performance. However, as all these approaches seek to indirectly model the intermediate geometric variables, they have inherent limitations in either training or inference process (see Sec. 2 for a detailed description). Therefore, an ideal solution would still be directly modeling the atomic coordinates and at the same time taking the roto-translational invariance property into account.

In this paper, we propose such a solution called GEODIFF, a principled probabilistic framework based on denoising diffusion models (Sohl-Dickstein et al., 2015). Our approach is inspired by the *diffusion process* in nonequilibrium thermodynamics (De Groot & Mazur, 2013). We view atoms as particles in a thermodynamic system, which gradually diffuse from the original states to a noisy distribution in contact with a heat bath. At each time step, stochastic noises are added to the atomic positions. Our high-level idea is learning to reverse the diffusion process, which recovers the target geometric distribution from the noisy distribution. In particular, inspired by recent progress of denoising diffusion models on image generation (Ho et al., 2020; Song et al., 2020), we view the noisy geometries at different timesteps as latent variables, and formulate both the forward diffusion and reverse denoising process as Markov chains. Our goal is to learn the transition kernels such that the reverse process can recover realistic conformations from the chaotic positions sampled from a noise distribution. However, extending existing methods to geometric generation is highly non-trivial: a direct application of diffusion models on the conformation generation task lead to poor generation quality. As mentioned above, molecular conformations are roto-translational invariant, *i.e.*, the estimated (conditional) likelihood should be unaffected by translational and rotational transformations (Köhler et al., 2020). To this end, we first theoretically show that a Markov process starting from an roto-translational *invariant* prior distribution and evolving with roto-translational *equivariant* Markov kernels can induce an roto-translational *invariant* density function. We further provide practical parameterization to define a roto-translational *invariant* prior distribution and a Markov kernel imposing the equivariance constraints. In addition, we derive a weighted variational lower bound of the conditional likelihood of molecular conformations, which also enjoys the roto-translational invariance and can be efficiently optimized.

A unique strength of GEODIFF is that it directly acts on the atomic coordinates and entirely bypasses the usage of intermediate elements for both training and inference. This general formulation enjoys several crucial advantages. First, the model can be naturally trained end-to-end without involving any sophisticated techniques like bilevel programming (Xu et al., 2021b), which benefits from small optimization variances. Besides, instead of solving geometries from bond lengths or angles, the one-stage sampling fashion avoids accumulating any intermediate error, and therefore leads to more accurate predicted structures. Moreover, GEODIFF enjoys a high model capacity to approximate the complex distribution of conformations. Thus, the model can better estimate the highly multi-modal distribution and generate structures with high quality and diversity.

We conduct comprehensive experiments on multiple benchmarks, including conformation generation and property prediction tasks. Numerical results show that GEODIFF consistently outperforms existing state-of-the-art machine learning approaches, and by a large margin on the more challenging large molecules. The significantly superior performance demonstrate the high capacity to model the complex distribution of molecular conformations and generate both diverse and accurate molecules.

## 2 RELATED WORK

Recently, various deep generative models have been proposed for conformation generation. Among them, CVGAE (Mansimov et al., 2019) first proposed a VAE model to directly generate 3D atomic coordinates, which fails to preserve the roto-translation equivariance property of conformations and suffers from poor performance. To address this problem, the majority of subsequent models are based on intermediate geometric elements such as atomic distances and torsion angles. A favorable property of these elements is the roto-translational invariance, (*e.g.* atomic distances does not change when rotating the molecule), which has been shown to be an important inductive bias for molecular geometry modeling (Köhler et al., 2020). However, such a decomposition suffers from several drawbacks for either training or sampling. For example, GRAPHDG (Simm & Hernandez-Lobato, 2020) and CGCF (Xu et al., 2021a) proposed to predict the interatomic distance matrix by VAE and Flow respectively, and then solve the geometry through the Distance Geometry (DG) technique (Liberti et al., 2014), which searches reasonable coordinates that matches with the predicted

distances. CONFVAE further improves this pipeline by designing an end-to-end framework via bilevel optimization (Xu et al., 2021b). However, all these approaches suffer from the accumulated error problem, meaning that the noise in the predicted distances will misguide the coordinate searching process and lead to inaccurate or even erroneous structures. To overcome this problem, CONFGF (Shi et al., 2021; Luo et al., 2021) proposed to learn the gradient of the log-likelihood *w.r.t* coordinates. However, in practice the model is still aided by intermediate geometric elements, in that it first estimates the gradient *w.r.t* interatomic distances via denoising score matching (DSM) (Song & Ermon, 2019; 2020), and then derives the gradient of coordinates using the chain rule. The problem is, by learning the distance gradient via DSM, the model is fed with perturbed distance matrices, which may violate the triangular inequality or even contain negative values. As a consequence, the model is actually learned over invalid distance matrices but tested with valid ones calculated from coordinates, making it suffer from serious out-of-distribution (Hendrycks & Gimpel, 2016) problem. Most recently, another concurrent work (Ganea et al., 2021) proposed a highly *systematic* (rule-based) pipeline named GEOMOL, which learns to predict a minimal set of geometric quantities (*i.e.* length and angles) and then reconstruct the local and global structures of the conformation in a sophisticated procedure. Besides, there has also been efforts to use reinforcement learning for conformation search Gogineni et al. (2020). Nevertheless, this method relies on rigid rotor approximation and can only model the torsion angles, and thus fundamentally differs from other approaches.

## 3 PRELIMINARIES

### 3.1 NOTATIONS AND PROBLEM DEFINITION

**Notations.** In this paper each molecule with $n$ atoms is represented as an undirected graph $\mathcal{G} = \langle \mathcal{V}, \mathcal{E} \rangle$, where $\mathcal{V} = \{v_i\}_{i=1}^n$ is the set of vertices representing atoms and $\mathcal{E} = \{e_{ij} \mid (i,j) \subseteq |\mathcal{V}| \times |\mathcal{V}|\}$ is the set of edges representing inter-atomic bonds. Each node $v_i \in \mathcal{V}$ describes the atomic attributes, *e.g.*, the element type. Each edge $e_{ij} \in \mathcal{E}$ describes the corresponding connection between $v_i$ and $v_j$, and is labeled with its chemical type. In addition, we also assign the unconnected edges with a *virtual* type. For the geometry, each atom in $\mathcal{V}$ is embedded by a coordinate vector $\boldsymbol{c} \in \mathbb{R}^3$ into the 3-dimensional space, and the full set of positions (*i.e.*, the conformation) can be represented as a matrix $\mathcal{C} = [\boldsymbol{c}_1, \boldsymbol{c}_2, \cdots, \boldsymbol{c}_n] \in \mathbb{R}^{n \times 3}$.

**Problem Definition.** The task of *molecular conformation generation* is a conditional generative problem, where we are interested in generating stable conformations for a provided graph $\mathcal{G}$. Given multiple graphs $\mathcal{G}$, and for each $\mathcal{G}$ given its conformations $\mathcal{C}$ as *i.i.d* samples from an underlying Boltzmann distribution (Noé et al., 2019), our goal is learning a generative model $p_\theta(\mathcal{C}|\mathcal{G})$, which is easy to draw samples from, to approximate the Boltzmann function.

### 3.2 EQUIVARIANCE

*Equivariance* is ubiquitous in machine learning for atomic systems, *e.g.*, the vectors of atomic dipoles or forces should rotate accordingly *w.r.t.* the conformation coordinates (Thomas et al., 2018; Weiler et al., 2018; Fuchs et al., 2020; Miller et al., 2020; Simm et al., 2021; Batzner et al., 2021). It has shown effectiveness to integrate such inductive bias into model parameterization for modeling 3D geometry, which is critical for the generalization capacity (Köhler et al., 2020; Satorras et al., 2021a). Formally, a function $\mathcal{F} : \mathcal{X} \to \mathcal{Y}$ is equivariant *w.r.t* a group $G$ if:

$$\mathcal{F} \circ T_g(x) = S_g \circ \mathcal{F}(x), \tag{1}$$

where $T_g$ and $S_g$ are transformations for an element $g \in G$, acting on the vector spaces $\mathcal{X}$ and $\mathcal{Y}$, respectively. In this work, we consider the SE(3) group, *i.e.*, the group of rotation, translation in 3D space. This requires the estimated likelihood unaffected with translational and rotational transformations, and we will elaborate on how our method satisfy this property in Sec. 4.

## 4 GEODIFF METHOD

In this section, we elaborate on the proposed equivariant diffusion framework. We first present a high level description of our 3D diffusion formulation in Sec. 4.1, based on recent progress of denoising diffusion models (Sohl-Dickstein et al., 2015; Ho et al., 2020). Then we emphasize several

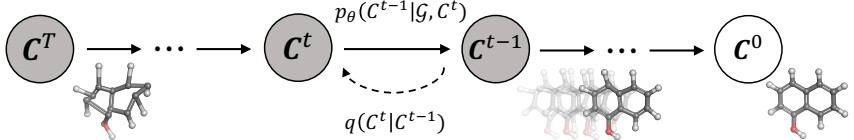

Figure 1: Illustration of the diffusion and reverse process of GEODIFF. For diffusion process, noise from fixed posterior distributions $q(\mathcal{C}^t|\mathcal{C}^{t-1})$ is gradually added until the conformation is destroyed. Symmetrically, for generative process, an initial state $\mathcal{C}^T$ is sampled from standard Gaussian distribution, and the conformation is progressively refined via the Markov kernels $p_\theta(\mathcal{C}^{t-1}|\mathcal{G}, \mathcal{C}^t)$.

non-trivial challenges of building diffusion models for geometry generation scenario, and show how we technically tackle these issues. Specifically, in Sec. 4.2, we present how we parameterize $p_\theta(\mathcal{C}|\mathcal{G})$ so that the conditional likelihood is roto-translational invariant, and in Sec. 4.3, we introduce our surgery of the training objective to make the optimization also invariant of translation and rotation. Finally, we briefly show how to draw samples from our model in Sec. 4.4.

## 4.1 FORMULATION

Let $\mathcal{C}^0$ denotes the ground truth conformations and let $\mathcal{C}^t$ for $t = 1, \cdots, T$ be a sequence of latent variables with the same dimension, where $t$ is the index for diffusion steps. Then a diffusion probabilistic model (Sohl-Dickstein et al., 2015) can be described as a latent variable model with two processes: the forward *diffusion* process, and the reverse *generative* process. Intuitively, the *diffusion process* progressively injects small noises to the data $\mathcal{C}^0$, while the *generative process* learns to revert the diffusion process by gradually eliminating the noise to recover the ground truth. We provide a high-level schematic of the processes in Fig. 1.

**Diffusion process.** Following the physical insight, we model the particles $\mathcal{C}$ as an evolving thermodynamic system. With time going by, the equilibrium conformation $\mathcal{C}^0$ will gradually diffuse to the next chaotic states $\mathcal{C}^t$, and finally converge into a white noise distribution after $T$ iterations. Different from typical latent variable models, in diffusion model this *forward process* is defined as a fixed (rather than trainable) posterior distribution $q(\mathcal{C}^{1:T}|\mathcal{C}^0)$. Specifically, we define it as a Markov chain according to a fixed variance schedule $\beta_1, \ldots, \beta_T$:

$$q(\mathcal{C}^{1:T}|\mathcal{C}^0) = \prod_{t=1}^{T} q(\mathcal{C}^t|\mathcal{C}^{t-1}), \quad q(\mathcal{C}^t|\mathcal{C}^{t-1}) = \mathcal{N}(\mathcal{C}^t; \sqrt{1-\beta_t}\mathcal{C}^{t-1}, \beta_t I). \quad (2)$$

Note that, in this work we do not impose specific (invariance) requirement upon the diffusion process, as long as it can efficiently draw noisy samples for training the generative process $p_\theta(\mathcal{C}^0)$.

Let $\alpha_t = 1 - \beta_t$ and $\bar{\alpha}_t = \prod_{s=1}^{t} \alpha_s$, a special property of the forward process is that $q(\mathcal{C}^t|\mathcal{C}^0)$ of arbitrary timestep $t$ can be calculated in closed form $q(\mathcal{C}^t|\mathcal{C}^0) = \mathcal{N}(\mathcal{C}^t; \sqrt{\bar{\alpha}_t}\mathcal{C}^0, (1-\bar{\alpha}_t)I)$[2]. This indicates with sufficiently large $T$, the whole forward process will convert $\mathcal{C}^0$ to whitened isotropic Gaussian, and thus it is natural to set $p(\mathcal{C}^T)$ as a standard Gaussian distribution.

**Reverse Process.** Our goal is learning to recover conformations $\mathcal{C}^0$ from the white noise $\mathcal{C}^T$, given specified molecular graphs $\mathcal{G}$. We consider this generative procedure as a reverse dynamics of the above diffusion process, starting from the noisy particles $\mathcal{C}^T \sim p(\mathcal{C}^T)$. We formulate this reverse dynamics as a conditional Markov chain with learnable transitions:

$$p_\theta(\mathcal{C}^{0:T-1}|\mathcal{G}, \mathcal{C}^T) = \prod_{t=1}^{T} p_\theta(\mathcal{C}^{t-1}|\mathcal{G}, \mathcal{C}^t), \quad p_\theta(\mathcal{C}^{t-1}|\mathcal{G}, \mathcal{C}^t) = \mathcal{N}(\mathcal{C}^{t-1}; \mu_\theta(\mathcal{G}, \mathcal{C}^t, t), \sigma_t^2 I). \quad (3)$$

Herein $\mu_\theta$ are parameterized neural networks to estimate the means, and $\sigma_t$ can be any user-defined variance. The initial distribution $p(\mathcal{C}^T)$ is set as a standard Gaussian. Given a graph $\mathcal{G}$, its 3D structure is generated by first drawing chaotic particles $\mathcal{C}^T$ from $p(\mathcal{C}^T)$, and then iteratively refined through the reverse Markov kernels $p_\theta(\mathcal{C}^{t-1}|\mathcal{G}, \mathcal{C}^t)$.

---

[2]Detailed derivations are provided in the Appendix A.

Having formulated the *reverse* dynamics, the marginal likelihood can be calculated by $p_\theta(\mathcal{C}^0|\mathcal{G}) = \int p(\mathcal{C}^T)p_\theta(\mathcal{C}^{0:T-1}|\mathcal{G},\mathcal{C}^T)\mathrm{d}\boldsymbol{\mathcal{C}}^{1:T}$. Herein a non-trivial problem is that the likelihood should be invariant *w.r.t* translation and rotation, which has proved to be a critical inductive bias for 3D object generation (Köhler et al., 2020; Satorras et al., 2021a). In the following subsections, we will elaborate on how we parameterize the Markov kernels $p_\theta(\mathcal{C}^{t-1}|\mathcal{G},\mathcal{C}^t)$ to achieve this desired property, and also how to maximize this likelihood by taking the invariance into account.

## 4.2 EQUIVARIANT REVERSE GENERATIVE PROCESS

Instead of directly leveraging existing methods, we consider building the density $p_\theta(\mathcal{C}^0)$ that is invariant to rotation and translation transformations. Intuitively, this requires the likelihood to be unaffected by translations and rotations. Formally, let $T_g$ be some roto-translational transformations of a group element $g \in \mathrm{SE}(3)$, then we have the following statement:

**Proposition 1.** *Let $p(x_T)$ be an SE(3)-invariant density function, i.e., $p(x_T) = p(T_g(x_T))$. If Markov transitions $p(x_{t-1}|x_t)$ are SE(3)-equivariant, i.e., $p(x_{t-1}|x_t) = p(T_g(x_{t-1})|T_g(x_t))$, then we have that the density $p_\theta(x_0) = \int p(x_T)p_\theta(x_{0:T-1}|x_T)\mathrm{d}\boldsymbol{x}_{1:T}$ is also SE(3)-invariant.*

This proposition indicates that the dynamics starting from an invariant standard density along an equivariant Gaussian Markov kernel can result in an invariant density. Now we provide a practical implementation of GEODIFF based on the recent *denoising diffusion* framework (Ho et al., 2020).

**Invariant Initial Density** $p(\mathcal{C}^T)$**.** We first introduce the invariant distribution $p(\mathcal{C}^T)$, which will also be employed in the equivariant Markov chain. We borrow the idea from Köhler et al. (2020) to consider systems with zero center of mass (CoM), termed CoM-free systems. We define $p(\mathcal{C}^T)$ as a "CoM-free standard density" $\hat{\rho}(\mathcal{C})$, built upon an isotropic normal density $\rho(\mathcal{C})$: for evaluating the likelihood $\hat{\rho}(\mathcal{C})$ we can firstly translate $\mathcal{C}$ to zero CoM and then calculate $\rho(\mathcal{C})$, and for sampling from $\hat{\rho}(\mathcal{C})$ we can first sample from $\rho(\mathcal{C})$ and then move the CoM to zero.

We provide a formal theoretical analysis of $\hat{\rho}(\mathcal{C})$ in Appendix A. Intuitively, the isotropic Gaussian is manifestly invariant to rotations around the zero CoM. And by considering CoM-free system, moving the particles to zero CoM can always ensure the translational invariance. Consequently, $\hat{\rho}(\mathcal{C})$ is constructed as a roto-transitional invariant density.

**Equivariant Markov Kernels** $p(\mathcal{C}^{t-1}|\mathcal{G},\mathcal{C}^t)$**.** Similar to the prior density, we also consider equipping all intermediate structures $\mathcal{C}^t$ as CoM-free systems. Specifically, given mean $\mu_\theta(\mathcal{G},\mathcal{C}^t,t)$ and variance $\sigma_t$, the likelihood of $\mathcal{C}^{t-1}$ will be calculated by $\hat{\rho}(\frac{\mathcal{C}^{t-1}-\mu_\theta(\mathcal{G},\mathcal{C}^t,t)}{\sigma_t})$. The CoM-free Gaussian ensures the translation invariance in the Markov kernels. Consequently, to achieve the equivariant property defined in Proposition 1, we focus on the rotation equivariance.

Then in general, the key requirement is to ensure the means $\mu_\theta(\mathcal{G},\mathcal{C}^t,t)$ to be roto-translation equivariant *w.r.t* $\mathcal{C}^t$. Following Ho et al. (2020), we consider the following parameterization of $\mu_\theta$:

$$\mu_\theta(\mathcal{C}^t,t) = \frac{1}{\sqrt{\alpha_t}}\left(\mathcal{C}^t - \frac{\beta_t}{\sqrt{1-\bar{\alpha}_t}}\epsilon_\theta(\mathcal{G},\mathcal{C}^t,t)\right), \tag{4}$$

where $\epsilon_\theta$ are neural networks with trainable parameters $\theta$. Intuitively, the model $\epsilon_\theta$ learns to predict the noise necessary to decorrupt the conformations. This is analogous to the physical force fields (Schütt et al., 2017; Zhang et al., 2018; Hu et al., 2021; Shuaibi et al., 2021), which also gradually push particles towards convergence around the equilibrium states.

Now the problem is transformed to constructing $\epsilon_\theta$ to be roto-translational equivariant. We draw inspirations from recent equivariant networks (Thomas et al., 2018; Satorras et al., 2021b) to design an equivariant convolutional layer, named graph field network (GFN). In the $l$-th layer, GFN takes node embeddings $\mathbf{h}^l \in \mathbb{R}^{n \times b}$ ($b$ denotes the feature dimension) and corresponding coordinate embeddings $\mathbf{x}^l \in \mathbb{R}^{n \times 3}$ as inputs, and outputs $\mathbf{h}^{l+1}$ and $\mathbf{x}^{l+1}$ as follows:

$$\mathbf{m}_{ij} = \Phi_m\left(\mathbf{h}_i^l, \mathbf{h}_j^l, \|\mathbf{x}_i^l - \mathbf{x}_j^l\|^2, e_{ij}; \theta_m\right) \tag{5}$$

$$\mathbf{h}_i^{l+1} = \Phi_h\left(\mathbf{h}_i^l, \sum_{j\in\mathcal{N}(i)}\mathbf{m}_{ij}; \theta_h\right) \tag{6}$$

$$\mathbf{x}_i^{l+1} = \sum_{j\in\mathcal{N}(i)}\frac{1}{d_{ij}}\left(\mathbf{c}_i - \mathbf{c}_j\right)\Phi_x\left(\mathbf{m}_{ij}; \theta_x\right) \tag{7}$$

where $\Phi$ are feed-forward networks and $d_{ij}$ denotes interatomic distances. $\mathcal{N}(i)$ denotes the neighborhood of $i^{th}$ node, including both connected atoms and other ones within a radius threshold $\tau$, which enables the model to explicitly capture long-range interactions and support molecular graphs with disconnected components. Initial embeddings $\mathbf{h}^0$ are combinations of atom and timestep embeddings, and $\mathbf{x}^0$ are atomic coordinates. The main difference between proposed GFN and other GNNs lies in equation 7, where $\mathbf{x}$ is updated as a combination of radial directions weighted by $\Phi_x : \mathbb{R}^b \to \mathbb{R}$. Such vector field $\mathbf{x}^L$ enjoys the roto-translation equivariance property. Formally, we have:

**Proposition 2.** *Parameterizing $\epsilon_\theta(\mathcal{G}, \mathcal{C}, t)$ as a composition of $L$ GFN layers, and take the $\mathbf{x}^L$ after $L$ updates as the output. Then the noise vector field $\epsilon_\theta$ is SE(3) equivariant w.r.t the 3D system $\mathcal{C}$.*

Intuitively, given $\mathbf{h}^l$ already invariant and $\mathbf{x}^l$ equivariant, the message embedding $\mathbf{m}$ will also be invariant since it only depends on invariant features. Since $\mathbf{x}$ is updated with the relative differences $\mathbf{c}_i - \mathbf{c}_j$ weighted by invariant features, it will be translation-invariant and rotation-equivariant. Then inductively, composing $\epsilon_\theta$ with $L$ GFN layers enables equivariance with $\mathcal{C}^t$. We provide the formal proof of equivariance properties in Appendix A.

### 4.3 Improved Training Objective

Having formulated the generative process and the model parameterization, now we consider the practical training objective for the reverse dynamics. Since directly optimizing the exact log-likelihood is intractable, we instead maximize the usual variational lower bound (ELBO)[3]:

$$
\mathbb{E}\left[\log p_\theta(\mathcal{C}^0|\mathcal{G})\right] = \mathbb{E}\left[\log \mathbb{E}_{q(\mathcal{C}^{1:T}|\mathcal{C}^0)} \frac{p_\theta(\mathcal{C}^{0:T}|\mathcal{G})}{q(\mathcal{C}^{1:T}|\mathcal{C}^0)}\right]
$$

$$
\geq -\mathbb{E}_q\left[\sum_{t=1}^{T} D_{\mathrm{KL}}(q(\mathcal{C}^{t-1}|\mathcal{C}^t, \mathcal{C}^0)\|p_\theta(\mathcal{C}^{t-1}|\mathcal{C}^t, \mathcal{G}))\right] := -\mathcal{L}_{\mathrm{ELBO}} \tag{8}
$$

where $q(\mathcal{C}^{t-1}|\mathcal{C}^t, \mathcal{C}^0)$ is analytically tractable as $\mathcal{N}(\frac{\sqrt{\bar{\alpha}_{t-1}}\beta_t}{1-\bar{\alpha}_t}\mathcal{C}^0 + \frac{\sqrt{\alpha_t}(1-\bar{\alpha}_{t-1})}{1-\bar{\alpha}_t}\mathcal{C}^t, \frac{1-\bar{\alpha}_{t-1}}{1-\bar{\alpha}_t}\beta_t)^3$. Most recently, Ho et al. (2020) showed that under the parameterization in equation 4, the ELBO of the diffusion model can be further simplified by calculating the KL divergences between Gaussians as weighted $\mathcal{L}_2$ distances between the means $\epsilon_\theta$ and $\epsilon^3$. Formally, we have:

**Proposition 3.** *(Ho et al., 2020) Under the parameterization in equation 4, we have:*

$$
\mathcal{L}_{\mathrm{ELBO}} = \sum_{t=1}^{T} \gamma_t \mathbb{E}_{\{\mathcal{C}^0, \mathcal{G}\}\sim q(\mathcal{C}^0, \mathcal{G}), \epsilon\sim\mathcal{N}(0, I)}\left[\|\epsilon - \epsilon_\theta(\mathcal{G}, \mathcal{C}^t, t)\|_2^2\right] \tag{9}
$$

*where $\mathcal{C}^t = \sqrt{\bar{\alpha}_t}\mathcal{C}^0 + \sqrt{1-\bar{\alpha}_t}\epsilon$. The weights $\gamma_t = \frac{\beta_t}{2\alpha_t(1-\bar{\alpha}_{t-1})}$ for $t > 1$, and $\gamma_1 = \frac{1}{2\alpha_1}$.*

The intuition of this objective is to independently sample chaotic conformations of different timesteps from $q(\mathcal{C}^{t-1}|\mathcal{C}^t, \mathcal{C}^0)$, and use $\epsilon_\theta$ to model the noise vector $\epsilon$. To yield a better empirical performance, Ho et al. (2020) suggests to set all weights $\gamma_t$ as 1, which is in line with the the objectives of recent noise conditional score networks (Song & Ermon, 2019; 2020).

As $\epsilon_\theta$ is designed to be equivariant, it is natural to require its supervision signal $\epsilon$ to be equivariant with $\mathcal{C}^t$. Note that once this is achieved, the ELBO will also become invariant. However, the $\epsilon$ in the forward diffusion process is not imposed with such equivariance, violating the above properties. Here we propose two approaches to obtain the modified noise vector $\hat{\epsilon}$, which, after replacing $\epsilon$ in the $\mathcal{L}_2$ distance calculation in equation 9, achieves the desired equivariance:

**Alignment approach**. Considering the fact that $\epsilon$ can be calculated by $\frac{\mathcal{C}^t - \sqrt{\bar{\alpha}_t}\mathcal{C}^0}{\sqrt{1-\bar{\alpha}_t}}$, we can first rotate and translate $\mathcal{C}^0$ to $\hat{\mathcal{C}}^0$ by aligning *w.r.t* $\mathcal{C}^t$, and then compute $\hat{\epsilon}$ as $\frac{\mathcal{C}^t - \sqrt{\bar{\alpha}_t}\hat{\mathcal{C}}^0}{\sqrt{1-\bar{\alpha}_t}}$. Since the aligned conformation $\hat{\mathcal{C}}^0$ is equivariant with $\mathcal{C}^t$, the processed $\hat{\epsilon}$ will also enjoy the equivariance. Specifically, the alignment is implemented by first translating $\mathcal{C}^0$ to the same CoM of $\mathcal{C}^t$ and then solve the optimal rotation matrix by Kabsch alignment algorithm (Kabsch, 1976).

---

[3]The detailed derivations and full proofs are provided in Appendix A.

**Chain-rule approach**. Another meaningful observation is that by reparameterizing the Gaussian distribution $q(\mathcal{C}^t|\mathcal{C}^0)$ as $\mathcal{C}^t = \sqrt{\bar{\alpha}_t}\mathcal{C}^0 + \sqrt{1-\bar{\alpha}_t}\epsilon$, $\epsilon$ can be viewed as a weighted score function $\sqrt{1-\bar{\alpha}_t}\nabla_{\mathcal{C}^t} q(\mathcal{C}^t|\mathcal{C}^0)$. Shi et al. (2021) recently shows that generally this score function $\nabla_{\mathcal{C}^t} q(\mathcal{C}^t|\cdot)$ can be designed to be equivariant by decomposing it into $\partial_{\mathcal{C}^t}\mathbf{d}^t \nabla_{\mathbf{d}^t} q(\mathcal{C}^t|\cdot)$ with the chain rule, where $\mathbf{d}^t$ can be any invariant features of the structures $\mathcal{C}^t$ such as the inter-atomic distances. We refer readers to Shi et al. (2021) for more details. The insight is that as gradient of invariant variables $w.r.t$ equivariant variables, the partial derivative $\partial_{\mathcal{C}^t}\mathbf{d}^t$ will always be equivalent with $\mathcal{C}^t$. In this work, under the common assumption that $\mathbf{d}$ also follows a Gaussian distribution (Kingma & Welling, 2013), our practical implementation is to first approximately calculate $\nabla_{\mathbf{d}^t} q(\mathcal{C}^t|\mathcal{C}^0)$ as $\frac{\mathbf{d}^t - \sqrt{\bar{\alpha}_t}\mathbf{d}^0}{1-\bar{\alpha}_t}$, and then compute the modified noise vector $\hat{\epsilon}$ as $\sqrt{1-\bar{\alpha}_t}\,\partial_{\mathcal{C}^t}\mathbf{d}^t\big(\frac{\mathbf{d}^t - \sqrt{\bar{\alpha}_t}\mathbf{d}^0}{1-\bar{\alpha}_t}\big) = \frac{\partial_{\mathcal{C}^t}\mathbf{d}^t \cdot (\mathbf{d}^t - \sqrt{\bar{\alpha}_t}\mathbf{d}^0)}{\sqrt{1-\bar{\alpha}_t}}$.

## 4.4 SAMPLING

With a learned reverse dynamics $\epsilon_\theta(\mathcal{G}, \mathcal{C}^t, t)$, the transition means $\mu_\theta(\mathcal{G}, \mathcal{C}^t, t)$ can be calculated by equation 4. Thus, given a graph $\mathcal{G}$, its geometry $\mathcal{C}^0$ is generated by first sampling chaotic particles $\mathcal{C}^T \sim p(\mathcal{C}^T)$, and then progressively sample $\mathcal{C}^{t-1} \sim p_\theta(\mathcal{C}^{t-1}|\mathcal{G}, \mathcal{C}^t)$ for $t = T, T-1, \cdots, 1$. This process is Markovian, which gradually shifts the previous noisy positions towards equilibrium states. We provide the pseudo code of the whole sampling process in Algorithm 1.

---

**Algorithm 1** Sampling Algorithm of GEODIFF.

---

**Input**: the molecular graph $\mathcal{G}$, the learned reverse model $\epsilon_\theta$.
**Output**: the molecular conformation $\mathcal{C}$.
1:  Sample $\mathcal{C}^T \sim p(\mathcal{C}^T) = \mathcal{N}(0, I)$
2:  **for** $s = T, T-1, \cdots, 1$ **do**
3:      Shift $\mathcal{C}^s$ to zero CoM
4:      Compute $\mu_\theta(\mathcal{C}^s, \mathcal{G}, s)$ from $\epsilon_\theta(\mathcal{C}^s, \mathcal{G}, s)$ using equation 4
5:      Sample $\mathcal{C}^{s-1} \sim \mathcal{N}(\mathcal{C}^{s-1}; \mu_\theta(\mathcal{C}^s, \mathcal{G}, s), \sigma_t^2 I)$
6:  **end for**
7:  **return** $\mathcal{C}^0$ as $\mathcal{C}$

---

## 5 EXPERIMENT

In this section, we empirically evaluate GEODIFF on the task of equilibrium conformation generation for both small and drug-like molecules. Following existing work (Shi et al., 2021; Ganea et al., 2021), we test the proposed method as well as the competitive baselines on two standard benchmarks: **Conformation Generation** (Sec. 5.2) and **Property Prediction** (Sec. 5.3). We first present the general experiment setups, and then describe task-specific evaluation protocols and discuss the results in each section. The implementation details are provided in Appendix C.

### 5.1 EXPERIMENT SETUP

**Datasets.** Following prior works (Xu et al., 2021a;b), we also use the recent GEOM-QM9 (Ramakrishnan et al., 2014) and GEOM-Drugs (Axelrod & Gomez-Bombarelli, 2020) datasets. The former one contains small molecules while the latter one are medium-sized organic compounds. We borrow the data split produced by Shi et al. (2021). For both datasets, the training split consists of $40,000$ molecules with $5$ conformations for each, resulting in $200,000$ conformations in total. The valid split share the same size as training split. The test split contains 200 distinct molecules, with $22,408$ conformations for QM9 and $14,324$ ones for Drugs.

**Baselines.** We compare GEODIFF with 6 recent or established state-of-the-art baselines. For the ML approaches, we test the following models with highest reported performance: CVGAE (Mansimov et al., 2019), GRAPHDG (Simm & Hernandez-Lobato, 2020), CGCF (Xu et al., 2021a), CONF-VAE (Xu et al., 2021b) and CONFGF (Shi et al., 2021). We also test the classic RDKIT (Riniker & Landrum, 2015) method, which is arguably the most popular open-source software for conformation generation. We refer readers to Sec. 2 for a detailed discussion of these models.

### 5.2 CONFORMATION GENERATION

**Evaluation metrics.** The task aims to measure both quality and diversity of generated conformations by different models. We follow Ganea et al. (2021) to evaluate 4 metrics built upon root-mean-square

Table 1: Results on the **GEOM-Drugs** dataset, without FF optimization.

| Models | COV-R (%) ↑ | | MAT-R (Å) ↓ | | COV-P (%) ↑ | | MAT-P (Å) ↓ | |
|---|---|---|---|---|---|---|---|---|
| | Mean | Median | Mean | Median | Mean | Median | Mean | Median |
| CVGAE | 0.00 | 0.00 | 3.0702 | 2.9937 | - | - | - | - |
| GRAPHDG | 8.27 | 0.00 | 1.9722 | 1.9845 | 2.08 | 0.00 | 2.4340 | 2.4100 |
| CGCF | 53.96 | 57.06 | 1.2487 | 1.2247 | 21.68 | 13.72 | 1.8571 | 1.8066 |
| CONFVAE | 55.20 | 59.43 | 1.2380 | 1.1417 | 22.96 | 14.05 | 1.8287 | 1.8159 |
| GEOMOL | 67.16 | 71.71 | 1.0875 | 1.0586 | - | - | - | - |
| CONFGF | 62.15 | 70.93 | 1.1629 | 1.1596 | 23.42 | 15.52 | 1.7219 | 1.6863 |
| **GEODIFF-A** | 88.36 | 96.09 | 0.8704 | 0.8628 | 60.14 | 61.25 | 1.1864 | 1.1391 |
| **GEODIFF-C** | **89.13** | **97.88** | **0.8629** | **0.8529** | **61.47** | **64.55** | **1.1712** | **1.1232** |

\* The COV-R and MAT-R results of CVGAE, GRAPHDG, CGCF, and CONFGF are borrowed from Shi et al. (2021). The results of GEOMOL are borrowed from a most recent study Zhu et al. (2022). Other results are obtained by our own experiments. The results of all models for the GEOM-QM9 dataset (summarized in Tab. 5) are collected in the same way.

deviation (RMSD), which is defined as the normalized Frobenius norm of two atomic coordinates matrices, after alignment by Kabsch algorithm (Kabsch, 1976). Formally, let $S_g$ and $S_r$ denote the sets of generated and reference conformers respectively, then the **Cov**erage and **Mat**ching metrics (Xu et al., 2021a) following the conventional *Recall* measurement can be defined as:

$$\text{COV-R}(S_g, S_r) = \frac{1}{|S_r|} \left| \left\{ \mathcal{C} \in S_r | \text{RMSD}(\mathcal{C}, \hat{\mathcal{C}}) \leq \delta, \hat{\mathcal{C}} \in S_g \right\} \right|, \quad (10)$$

$$\text{MAT-R}(S_g, S_r) = \frac{1}{|S_r|} \sum_{\mathcal{C} \in S_r} \min_{\hat{\mathcal{C}} \in S_g} \text{RMSD}(\mathcal{C}, \hat{\mathcal{C}}), \quad (11)$$

where $\delta$ is a pre-defined threshold. The other two metrics COV-P and MAT-P inspired by *Precision* can be defined similarly but with the generated and reference sets exchanged. In practice, $S_g$ is set as twice of the size of $S_r$ for each molecule. Intuitively, the COV scores measure the percentage of structures in one set covered by another set, where covering means the RMSD between two conformations is within a certain threshold $\delta$. By contrast, the MAT scores measure the average RMSD of conformers in one set with its closest neighbor in another set. In general, higher COV rates or lower MAT score suggest that more realistic conformations are generated. Besides, the *Precision* metrics depend more on the quality, while the *Recall* metrics concentrate more on the diversity. Either metrics can be more appealing considering the specific scenario. Following previous works (Xu et al., 2021a; Ganea et al., 2021), $\delta$ is set as 0.5Å and 1.25Å for QM9 and Drugs datasets respectively.

**Results & discussion.** The results are summarized in Tab. 1 and Tab. 5 (left in Appendix. D). As noted in Sec. 4.3, GEODIFF can be trained with two types of modified ELBO, named *alignment* and *chain-rule* approaches. We denote models learned by these two objectives as GEODIFF-A and GEODIFF-C respectively. As shown in the tables, GEODIFF consistently outperform the state-of-the-art ML models on all datasets and metrics, especially by a significant margin for more challenging large molecules (Drugs dataset). The results demonstrate the superior capacity of GEODIFF to model the multi modal distribution, and generative both accurate and diverse conformations. We also notice that in general GEODIFF-C performs slightly better than GEODIFF-A, which suggests that *chain-rule approach* leads to a better optimization procedure. We thus take GEODIFF-C as the representative in the following comparisons. We visualize samples generated by different models in Fig. 2 to provide a qualitative comparison, where GEODIFF is shown to capture better both local and global structures.

On the more challenging Drugs dataset, we further test RDKIT. As shown in Tab. 2, our observation is in line with previous studies (Shi et al., 2021) that the state-of-the-art ML models (shown in Tab. 1) perform better on COV-R and MAT-R. However, for the new *Precision*-based metrics we found that ML models are still not comparable. This indicates that ML models tend to explore more possible representatives while RDKIT concentrates on a few most common ones, prioritizes quality over diversity. Previous works (Mansimov et al., 2019; Xu et al., 2021b) suggest that this is because RDKIT involves an additional empirical force field (FF) (Halgren, 1996) to optimize the structure, and we follow them to also combine GEODIFF with FF to yield a more fair comparison. Results in

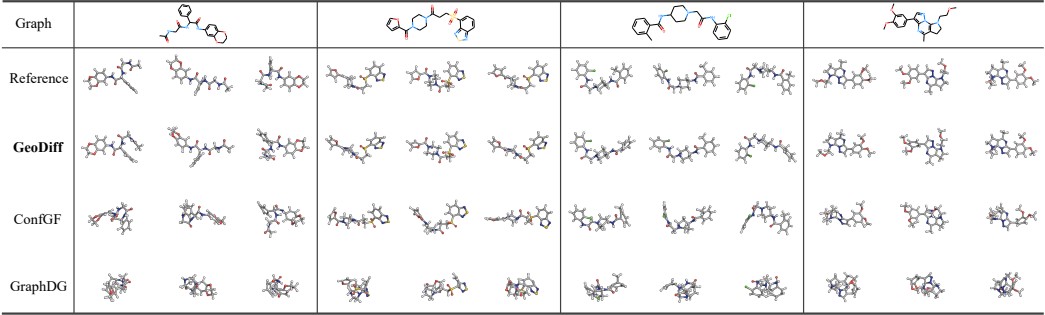

Figure 2: Examples of generated structures from Drugs dataset. For every model, we show the conformation best-aligned with the ground truth. More examples are provided in Appendix E.

Table 2: Results on the **GEOM-Drugs** dataset, with FF optimization.

| Models | COV-R (%) ↑ | | MAT-R (Å) ↓ | | COV-P (%) ↑ | | MAT-P (Å) ↓ | |
| | Mean | Median | Mean | Median | Mean | Median | Mean | Median |
| --- | --- | --- | --- | --- | --- | --- | --- | --- |
| RDKIT | 60.91 | 65.70 | 1.2026 | 1.1252 | 72.22 | 88.72 | 1.0976 | 0.9539 |
| **GEODIFF + FF** | **92.27** | **100.00** | **0.7618** | **0.7340** | **84.51** | **95.86** | **0.9834** | **0.9221** |

Tab. 2 demonstrate that GEODIFF +FF can keep the superior diversity (*Recall* metrics) while also enjoy significantly improved accuracy ((*Precision* metrics)).

## 5.3 PROPERTY PREDICTION

**Evaluation metrics.** This task estimates the molecular *ensemble properties* (Axelrod & Gomez-Bombarelli, 2020) over a set of generated conformations. This can provide an direct assessment on the quality of generated samples. In specific, we follow Shi et al. (2021) to extract a split from GEOM-QM9 covering 30 molecules, and generate 50 samples for each. Then we use

Table 3: MAE of predicted ensemble properties in eV.

| Method | $\overline{E}$ | $E_{min}$ | $\overline{\Delta\epsilon}$ | $\Delta\epsilon_{min}$ | $\Delta\epsilon_{max}$ |
| --- | --- | --- | --- | --- | --- |
| RDKIT | 0.9233 | 0.6585 | 0.3698 | 0.8021 | 0.2359 |
| GRAPHDG | 9.1027 | 0.8882 | 1.7973 | 4.1743 | 0.4776 |
| CGCF | 28.9661 | 2.8410 | 2.8356 | 10.6361 | 0.5954 |
| CONFVAE | 8.2080 | 0.6100 | 1.6080 | 3.9111 | 0.2429 |
| CONFGF | 2.7886 | 0.1765 | 0.4688 | 2.1843 | **0.1433** |
| GEODIFF | **0.25974** | **0.1551** | **0.3091** | **0.7033** | 0.1909 |

the chemical toolkit PSI4 (Smith et al., 2020) to calculate each conformer's energy $E$ and HOMO-LUMO gap $\epsilon$, and compare the average energy $\overline{E}$, lowest energy $E_{min}$, average gap $\overline{\Delta\epsilon}$, minimum gap $\Delta\epsilon_{min}$, and maximum gap $\Delta\epsilon_{max}$ with the ground truth.

**Results & discussions.** The mean absolute errors (MAE) between calculated properties and the ground truth are reported in Tab. 3. CVGAE is excluded due to the poor performance, which is also reported in Simm & Hernandez-Lobato (2020); Shi et al. (2021). The properties are highly sensitive to geometric structure, and thus the superior performance demonstrate that GEODIFF can consistently predict more accurate conformations across different molecules.

## 6 CONCLUSION

We propose GEODIFF, a novel probabilistic model for generating molecular conformations. GEODIFF marries denoising diffusion models with geometric representations, where we parameterize the reverse generative dynamics as a Markov chain, and novelly impose roto-translational invariance into the density with equivariant Markov kernels. We derive a tractable invariant objective from the variational lower bound to optimize the likelihood. Comprehensive experiments over multiple tasks demonstrate that GEODIFF is competitive with the existing state-of-the-art models. Future work includes further improving or accelerating the model with other recent progress of diffusion models, and extending our method to other challenging structures such as proteins.

A­CKNOWLEDGEMENT

Minkai thanks Huiyu Cai, David Wipf, Zuobai Zhang, and Zhaocheng Zhu for their helpful discussions and comments. This project is supported by the Natural Sciences and Engineering Research Council (NSERC) Discovery Grant, the Canada CIFAR AI Chair Program, collaboration grants between Microsoft Research and Mila, Samsung Electronics Co., Ltd., Amazon Faculty Research Award, Tencent AI Lab Rhino-Bird Gift Fund and a NRC Collaborative R&D Project (AI4D-CORE-06). This project was also partially funded by IVADO Fundamental Research Project grant PRF-2019-3583139727. The Stanford team is supported by NSF(#1651565, #1522054, #1733686), ONR (N000141912145), AFOSR (FA95501910024), ARO (W911NF-21-1-0125) and Sloan Fellowship.

R­EFERENCES

Mohammed AlQuraishi. End-to-end differentiable learning of protein structure. *Cell systems*, 8(4): 292–301, 2019.

Simon Axelrod and Rafael Gomez-Bombarelli. Geom: Energy-annotated molecular conformations for property prediction and molecular generation. *arXiv preprint arXiv:2006.05531*, 2020.

Simon Batzner, Tess E Smidt, Lixin Sun, Jonathan P Mailoa, Mordechai Kornbluth, Nicola Molinari, and Boris Kozinsky. Se (3)-equivariant graph neural networks for data-efficient and accurate interatomic potentials. *arXiv preprint arXiv:2101.03164*, 2021.

Julian Chibane, Thiemo Alldieck, and Gerard Pons-Moll. Implicit functions in feature space for 3d shape reconstruction and completion. In *Proceedings of the IEEE/CVF Conference on Computer Vision and Pattern Recognition*, pp. 6970–6981, 2020.

Sybren Ruurds De Groot and Peter Mazur. *Non-equilibrium thermodynamics*. Courier Corporation, 2013.

Laurent Dinh, Jascha Sohl-Dickstein, and Samy Bengio. Density estimation using Real NVP. In *ICLR*, 2017.

David K Duvenaud, Dougal Maclaurin, Jorge Iparraguirre, Rafael Bombarell, Timothy Hirzel, Alán Aspuru-Guzik, and Ryan P Adams. Convolutional networks on graphs for learning molecular fingerprints. In *Advances in neural information processing systems*, pp. 2224–2232, 2015.

Fabian Fuchs, Daniel Worrall, Volker Fischer, and Max Welling. Se(3)-transformers: 3d roto-translation equivariant attention networks. *NeurIPS*, 2020.

Octavian-Eugen Ganea, Lagnajit Pattanaik, Connor W Coley, Regina Barzilay, Klavs F Jensen, William H Green, and Tommi S Jaakkola. Geomol: Torsional geometric generation of molecular 3d conformer ensembles. *arXiv preprint arXiv:2106.07802*, 2021.

Niklas WA Gebauer, Michael Gastegger, Stefaan SP Hessmann, Klaus-Robert Müller, and Kristof T Schütt. Inverse design of 3d molecular structures with conditional generative neural networks. *arXiv preprint arXiv:2109.04824*, 2021.

Justin Gilmer, Samuel S Schoenholz, Patrick F Riley, Oriol Vinyals, and George E Dahl. Neural message passing for quantum chemistry. In *Proceedings of the 34th International Conference on Machine Learning-Volume 70*, pp. 1263–1272. JMLR. org, 2017.

T. Gogineni, Ziping Xu, Exequiel Punzalan, Runxuan Jiang, Joshua A Kammeraad, Ambuj Tewari, and P. Zimmerman. Torsionnet: A reinforcement learning approach to sequential conformer search. *ArXiv*, abs/2006.07078, 2020.

Thomas A Halgren. Merck molecular force field. v. extension of mmff94 using experimental data, additional computational data, and empirical rules. *Journal of Computational Chemistry*, 17(5-6): 616–641, 1996.

Paul CD Hawkins. Conformation generation: the state of the art. *Journal of Chemical Information and Modeling*, 57(8):1747–1756, 2017.

Dan Hendrycks and Kevin Gimpel. A baseline for detecting misclassified and out-of-distribution examples in neural networks. *arXiv preprint arXiv:1610.02136*, 2016.

Jonathan Ho, Ajay Jain, and Pieter Abbeel. Denoising diffusion probabilistic models. *arXiv preprint arXiv:2006.11239*, 2020.

Weihua Hu, Muhammed Shuaibi, Abhishek Das, Siddharth Goyal, Anuroop Sriram, Jure Leskovec, Devi Parikh, and Larry Zitnick. Forcenet: A graph neural network for large-scale quantum chemistry simulation. 2021.

John Ingraham, Adam J Riesselman, Chris Sander, and Debora S Marks. Learning protein structure with a differentiable simulator. In *International Conference on Learning Representations*, 2019.

Wengong Jin, Regina Barzilay, and Tommi Jaakkola. Junction tree variational autoencoder for molecular graph generation. *arXiv preprint arXiv:1802.04364*, 2018.

Bowen Jing, Stephan Eismann, Patricia Suriana, Raphael John Lamarre Townshend, and Ron Dror. Learning from protein structure with geometric vector perceptrons. In *International Conference on Learning Representations*, 2021.

John Jumper, Richard Evans, Alexander Pritzel, Tim Green, Michael Figurnov, Olaf Ronneberger, Kathryn Tunyasuvunakool, Russ Bates, Augustin Žídek, Anna Potapenko, et al. Highly accurate protein structure prediction with alphafold. *Nature*, 596(7873):583–589, 2021.

Wolfgang Kabsch. A solution for the best rotation to relate two sets of vectors. *Acta Crystallographica Section A: Crystal Physics, Diffraction, Theoretical and General Crystallography*, 32(5):922–923, 1976.

Diederik P. Kingma and Max Welling. Auto-encoding variational bayes. In *2nd International Conference on Learning Representations*, 2013.

Jonas Köhler, Leon Klein, and Frank Noe. Equivariant flows: Exact likelihood generative learning for symmetric densities. In *Proceedings of the 37th International Conference on Machine Learning*, 2020.

Leo Liberti, Carlile Lavor, Nelson Maculan, and Antonio Mucherino. Euclidean distance geometry and applications. *SIAM review*, 56(1):3–69, 2014.

Shitong Luo and Wei Hu. Diffusion probabilistic models for 3d point cloud generation. *ArXiv*, abs/2103.01458, 2021.

Shitong Luo, Chence Shi, Minkai Xu, and Jian Tang. Predicting molecular conformation via dynamic graph score matching. *Advances in Neural Information Processing Systems*, 34, 2021.

Elman Mansimov, Omar Mahmood, Seokho Kang, and Kyunghyun Cho. Molecular geometry prediction using a deep generative graph neural network. *arXiv preprint arXiv:1904.00314*, 2019.

B. Miller, M. Geiger, T. Smidt, and F. Noé. Relevance of rotationally equivariant convolutions for predicting molecular properties. *ArXiv*, abs/2008.08461, 2020.

Frank Noé, Simon Olsson, Jonas Köhler, and Hao Wu. Boltzmann generators: Sampling equilibrium states of many-body systems with deep learning. *Science*, 365(6457), 2019.

Raghunathan Ramakrishnan, Pavlo O Dral, Matthias Rupp, and O Anatole Von Lilienfeld. Quantum chemistry structures and properties of 134 kilo molecules. *Scientific data*, 1(1):1–7, 2014.

Sereina Riniker and Gregory A. Landrum. Better informed distance geometry: Using what we know to improve conformation generation. *Journal of Chemical Information and Modeling*, 55(12): 2562–2574, 2015.

Victor Garcia Satorras, Emiel Hoogeboom, Fabian B Fuchs, Ingmar Posner, and Max Welling. E (n) equivariant normalizing flows for molecule generation in 3d. *arXiv preprint arXiv:2105.09016*, 2021a.

Victor Garcia Satorras, Emiel Hoogeboom, and Max Welling. E(n) equivariant graph neural networks, 2021b.

Kristof Schütt, Pieter-Jan Kindermans, Huziel Enoc Sauceda Felix, Stefan Chmiela, Alexandre Tkatchenko, and Klaus-Robert Müller. Schnet: A continuous-filter convolutional neural network for modeling quantum interactions. In *Advances in Neural Information Processing Systems*, pp. 991–1001. Curran Associates, Inc., 2017.

Andrew W Senior, Richard Evans, John Jumper, James Kirkpatrick, Laurent Sifre, Tim Green, Chongli Qin, Augustin Žídek, Alexander WR Nelson, Alex Bridgland, et al. Improved protein structure prediction using potentials from deep learning. *Nature*, 577(7792):706–710, 2020.

Chence Shi, Minkai Xu, Zhaocheng Zhu, Weinan Zhang, Ming Zhang, and Jian Tang. Graphaf: a flow-based autoregressive model for molecular graph generation. *arXiv preprint arXiv:2001.09382*, 2020.

Chence Shi, Shitong Luo, Minkai Xu, and Jian Tang. Learning gradient fields for molecular conformation generation. *ArXiv*, 2021.

Muhammed Shuaibi, Adeesh Kolluru, Abhishek Das, Aditya Grover, Anuroop Sriram, Zachary Ulissi, and C Lawrence Zitnick. Rotation invariant graph neural networks using spin convolutions. *arXiv preprint arXiv:2106.09575*, 2021.

Gregor Simm and Jose Miguel Hernandez-Lobato. A generative model for molecular distance geometry. In Hal Daumé III and Aarti Singh (eds.), *Proceedings of the 37th International Conference on Machine Learning*, volume 119, pp. 8949–8958. PMLR, 2020.

Gregor N. C. Simm, Robert Pinsler, Gábor Csányi, and José Miguel Hernández-Lobato. Symmetry-aware actor-critic for 3d molecular design. In *International Conference on Learning Representations*, 2021.

Daniel G. A. Smith, L. Burns, A. Simmonett, R. Parrish, M. C. Schieber, Raimondas Galvelis, P. Kraus, H. Kruse, Roberto Di Remigio, Asem Alenaizan, A. M. James, S. Lehtola, Jonathon P Misiewicz, et al. Psi4 1.4: Open-source software for high-throughput quantum chemistry. *The Journal of chemical physics*, 2020.

Jascha Sohl-Dickstein, Eric A Weiss, Niru Maheswaranathan, and Surya Ganguli. Deep unsupervised learning using nonequilibrium thermodynamics. *arXiv preprint arXiv:1503.03585*, 2015.

Jiaming Song, Chenlin Meng, and Stefano Ermon. Denoising diffusion implicit models. *arXiv preprint arXiv:2010.02502*, 2020.

Yang Song and Stefano Ermon. Generative modeling by estimating gradients of the data distribution. In *Advances in Neural Information Processing Systems*, pp. 11918–11930, 2019.

Yang Song and Stefano Ermon. Improved techniques for training score-based generative models. *NeurIPS*, 2020.

N. Thomas, T. Smidt, Steven M. Kearnes, Lusann Yang, L. Li, Kai Kohlhoff, and P. Riley. Tensor field networks: Rotation- and translation-equivariant neural networks for 3d point clouds. *ArXiv*, 2018.

M. Weiler, M. Geiger, M. Welling, W. Boomsma, and T. Cohen. 3d steerable cnns: Learning rotationally equivariant features in volumetric data. In *NeurIPS*, 2018.

Minkai Xu, Shitong Luo, Yoshua Bengio, Jian Peng, and Jian Tang. Learning neural generative dynamics for molecular conformation generation. In *International Conference on Learning Representations*, 2021a.

Minkai Xu, Wujie Wang, Shitong Luo, Chence Shi, Yoshua Bengio, Rafael Gomez-Bombarelli, and Jian Tang. An end-to-end framework for molecular conformation generation via bilevel programming. *arXiv preprint arXiv:2105.07246*, 2021b.

Linfeng Zhang, Jiequn Han, Han Wang, Roberto Car, and Weinan E. Deep Potential Molecular Dynamics: A Scalable Model with the Accuracy of Quantum Mechanics. *Physical Review Letters*, 120(14):143001, 2018.

Jinhua Zhu, Yingce Xia, Chang Liu, Lijun Wu, Shufang Xie, Tong Wang, Yusong Wang, Wengang Zhou, Tao Qin, Houqiang Li, et al. Direct molecular conformation generation. *arXiv preprint arXiv:2202.01356*, 2022.

# A   PROOFS

## A.1   PROPERTIES OF THE DIFFUSION MODEL

We include proofs for several key properties of the probabilistic diffusion model here to be self-contained. For more detailed discussions, please refer to Ho et al. (2020). Let $\{\beta_0, ..., \beta_T\}$ be a sequence of variances, and $\alpha_t = 1 - \beta_t$ and $\bar{\alpha}_t = \prod_{s=1}^{t} \alpha_s$. The two following properties are crucial for deriving the final tractable objective in equation 9.

**Property 1.** *Tractable marginal of the forward process:*

$$q(\mathcal{C}^t|\mathcal{C}^0) = \int q(\mathcal{C}^{1:t}|\mathcal{C}^0)\, d\mathcal{C}^{1:(t-1)} = \mathcal{N}(\mathcal{C}^t;\ \sqrt{\bar{\alpha}_t}\mathcal{C}^0, (1-\bar{\alpha}_t)I).$$

*Proof.* Let $\epsilon_i$'s be independent standard Gaussian random variables. Then, by definition of the Markov kernels $q(\mathcal{C}^t|\mathcal{C}^{t-1})$ in equation 2, we have

$$\begin{aligned}
\mathcal{C}^t &= \sqrt{\alpha_t}\mathcal{C}^{t-1} + \sqrt{\beta_t}\epsilon_t \\
&= \sqrt{\alpha_t\alpha_{t-1}}\mathcal{C}^{t-2} + \sqrt{\alpha_t\beta_{t-1}}\epsilon_{t-1} + \sqrt{\beta_t}\epsilon_t \\
&= \sqrt{\alpha_t\alpha_{t-1}\alpha_{t-1}}\mathcal{C}^{t-3} + \sqrt{\alpha_t\alpha_{t-1}\beta_{t-2}}\epsilon_{t-2} + \sqrt{\alpha_t\beta_{t-1}}\epsilon_{t-1} + \sqrt{\beta_t}\epsilon_t \\
&= \cdots \\
&= \sqrt{\bar{\alpha}_t}\mathcal{C}^0 + \sqrt{\alpha_t\alpha_{t-1}\cdots\alpha_2\beta_1}\epsilon_1 + \cdots + \sqrt{\alpha_t\beta_{t-1}}\epsilon_{t-1} + \sqrt{\beta_t}\epsilon_t
\end{aligned} \tag{12}$$

Therefore $q(\mathcal{C}^t|\mathcal{C}^0)$ is still Gaussian, and the mean of $\mathcal{C}^t$ is $\sqrt{\bar{\alpha}_t}\mathcal{C}^0$, and the variance matrix is $(\alpha_t\alpha_{t-1}\cdots\alpha_2\beta_1 + \cdots + \alpha_t\beta_{t-1} + \beta_t)I = (1-\bar{\alpha}_t)I$. Then we have:

$$q(\mathcal{C}^t|\mathcal{C}^0) = \mathcal{N}(\mathcal{C}^t;\ \sqrt{\bar{\alpha}_t}\mathcal{C}^0,\ (1-\bar{\alpha}_t)I).$$

This property provides convenient closed-form evaluation of $\mathcal{C}^t$ knowing $\mathcal{C}^0$:

$$\mathcal{C}^t = \sqrt{\bar{\alpha}_t}\mathcal{C}^0 + \sqrt{1-\bar{\alpha}_t}\boldsymbol{\epsilon},$$

where $\boldsymbol{\epsilon} \sim \mathcal{N}(0, I)$.

Besides, it is worth noting that,

$$q(\mathcal{C}^T|\mathcal{C}^0) = \mathcal{N}(\mathcal{C}^T;\ \sqrt{\bar{\alpha}_T}\mathcal{C}^0,\ (1-\bar{\alpha}_T)I),$$

where $\bar{\alpha}_T = \prod_{t=1}^{T}(1-\beta_t)$ approaches zero with large $T$, which indicates the diffusion process can finally converge into a whitened noisy distribution. $\square$

**Property 2.** *Tractable posterior of the forward process:*

$$q(\mathcal{C}^{t-1}|\mathcal{C}^t, \mathcal{C}^0) = \mathcal{N}(\mathcal{C}^{t-1};\ \frac{\sqrt{\bar{\alpha}_{t-1}}\beta_t}{1-\bar{\alpha}_t}\mathcal{C}^0 + \frac{\sqrt{\alpha_t}(1-\bar{\alpha}_{t-1})}{1-\bar{\alpha}_t}\mathcal{C}^t, \frac{(1-\bar{\alpha}_{t-1})}{1-\bar{\alpha}_t}\beta_t I).$$

*Proof.* Let $\tilde{\beta}_t = \frac{1-\bar{\alpha}_{t-1}}{1-\bar{\alpha}_t}\beta_t$, then we can derive the posterior by Bayes rule:

$$\begin{aligned}
q(\mathcal{C}^{t-1}|\mathcal{C}^t, \mathcal{C}^0) \quad &= \frac{q(\mathcal{C}^t|\mathcal{C}^{t-1})\, q(\mathcal{C}^{t-1}|\mathcal{C}^0)}{q(\mathcal{C}^t|\mathcal{C}^0)} \\
&= \frac{\mathcal{N}(\mathcal{C}^t; \sqrt{\alpha_t}\mathcal{C}^{t-1}, \beta_t I)\, \mathcal{N}(\mathcal{C}^{t-1}; \sqrt{\bar{\alpha}_{t-1}}\mathcal{C}^0, (1-\bar{\alpha}_{t-1})I)}{\mathcal{N}(\mathcal{C}^t; \sqrt{\bar{\alpha}_t}\mathcal{C}^0, (1-\bar{\alpha}_t)I)} \\
&= (2\pi\beta_t)^{-\frac{d}{2}}(2\pi(1-\bar{\alpha}_{t-1}))^{-\frac{d}{2}}(2\pi(1-\bar{\alpha}_t))^{\frac{d}{2}} \times \\
&\quad \exp\left(-\frac{\|\mathcal{C}^t - \sqrt{\alpha_t}\mathcal{C}^{t-1}\|^2}{2\beta_t} - \frac{\|\mathcal{C}^{t-1} - \sqrt{\bar{\alpha}_{t-1}}\mathcal{C}^0\|^2}{2(1-\bar{\alpha}_{t-1})} + \frac{\|\mathcal{C}^t - \sqrt{\bar{\alpha}_t}\mathcal{C}^0\|^2}{2(1-\bar{\alpha}_t)}\right) \\
&= (2\pi\tilde{\beta}_t)^{-\frac{d}{2}}\exp\left(-\frac{1}{2\tilde{\beta}_t}\left\|\mathcal{C}^{t-1} - \frac{\sqrt{\bar{\alpha}_{t-1}}\beta_t}{1-\bar{\alpha}_t}\mathcal{C}^0 - \frac{\sqrt{\alpha_t}(1-\bar{\alpha}_{t-1})}{1-\bar{\alpha}_t}\mathcal{C}^t\right\|^2\right)
\end{aligned} \tag{13}$$

Then we have the posterior $q(\mathcal{C}^{t-1}|\mathcal{C}^t, \mathcal{C}^0)$ as the given form. $\square$

## A.2 PROOF OF PROPOSITION 1

Let $T_g$ be some roto-translational transformations of a group element $g \in$ SE(3), and let $p(x_T)$ be a density which is SE(3)-invariant, *i.e.*, $p(x_T) = p(T_g(x_T))$. If the Markov transitions $p(x_{t-1}|x_t)$ are SE(3)-equivariant, *i.e.*, $p(x_{t-1}|x_t) = p(T_g(x_{t-1})|T_g(x_t))$, then we have that the density $p_\theta(x_0) = \int p(x_T)p_\theta(x_{0:T-1}|x_T)\mathrm{d}\boldsymbol{x}_{1:T}$ is also SE(3)-invariant.

*Proof.*

$$
\begin{aligned}
p_\theta(T_g(x_0)) &= \int p(T_g(x_T))p_\theta(T_g(x_{0:T-1})|T_g(x_T))\mathrm{d}\boldsymbol{x}_{1:T} \\
&= \int p(T_g(x_T))\Pi_{t=1}^T p_\theta(T_g(x_{t-1})|T_g(x_t))\mathrm{d}\boldsymbol{x}_{1:T} \\
&= \int p(x_T)\Pi_{t=1}^T p_\theta(T_g(x_{t-1})|T_g(x_t))\mathrm{d}\boldsymbol{x}_{1:T} \quad \text{(invariant prior } p(x_T)) \\
&= \int p(x_T)\Pi_{t=1}^T p_\theta(x_{t-1}|x_t)\mathrm{d}\boldsymbol{x}_{1:T} \quad \text{(equivariant kernels } p(x_{t-1}|x_t)) \\
&= \int p(x_T)p_\theta(x_{0:T-1}|x_T)\mathrm{d}\boldsymbol{x}_{1:T} \\
&= p_\theta(x_0)
\end{aligned}
\tag{14}
$$

$\square$

## A.3 PROOF OF PROPOSITION 2

In this section we prove that the output $\mathbf{x}$ of GFN defined in equation 5, 6 and 7 is translationally invariant and rotationally equivariant with the input $\mathcal{C}$. Let $g \in \mathbb{R}^3$ denote any translation transformations and orthogonal matrices $R \in \mathbb{R}^{3\times3}$ denote any rotation transformations. let $R\mathbf{x}$ be shorthand for $(R\mathbf{x}_1, \cdots, R\mathbf{x}_N)$. Formally, we aim to prove that the model satisfies:

$$
R\mathbf{x}^{l+1}, \mathbf{h}^{l+1} = \text{GFN}(R\mathbf{x}^l, R\mathcal{C} + g, \mathbf{h}^l).
\tag{15}
$$

This equation indicates that, given $\mathbf{x}^l$ already rotationally equivalent with $\mathcal{C}$, and $\mathbf{h}^l$ already invariant, then such property can propagate through a single GFN layer to $\mathbf{x}^{l+1}$ and $\mathbf{h}^{l+1}$.

*Proof.* Firstly, given that $\mathbf{h}^l$ already invariant to SE(3) transformations, we have that the messages $\mathbf{m}_{ij}$ calculated from equation 5 will also be invariant. This is because it sorely relies on the distance between two atoms, which are manifestly invariant to rotations $\|R\mathbf{x}_i^l - R\mathbf{x}_j^l\|^2 = (\mathbf{x}_i^l - \mathbf{x}_j^l)^\top R^\top R(\mathbf{x}_i^l - \mathbf{x}_j^l) = (\mathbf{x}_i^l - \mathbf{x}_j^l)^\top I(\mathbf{x}_i^l - \mathbf{x}_j^l) = \|\mathbf{x}_i^l - \mathbf{x}_j^l\|^2$. Formally, the invariance of messages in equation 5 can be written as:

$$
\mathbf{m}_{i,j} = \Phi_m\left(\mathbf{h}_i^l, \mathbf{h}_j^l, \left\|R\mathbf{x}_i^l - R\mathbf{x}_j^l\right\|^2, e_{ij}\right) = \Phi_m\left(\mathbf{h}_i^l, \mathbf{h}_j^l, \left\|\mathbf{x}_i^l - \mathbf{x}_j^l\right\|^2, e_{ij}\right).
\tag{16}
$$

And similarly, the $\mathbf{h}^{t+1}$ updated from equation 6 will also be invariant.

Next, we prove that the vector $\mathbf{x}$ updated from equation 7 preserves rotational equivariance and translational invariance. Given $\mathbf{m}_{ij}$ already invariant as proven above, we have that:

$$
\sum_{j \in \mathcal{N}(i)} \frac{1}{d_{ij}}\left(R\mathbf{c}_i + g - R\mathbf{c}_j - g\right)\Phi_x\left(\mathbf{m}_{i,j}\right) = R\sum_{j \in \mathcal{N}(i)} \frac{1}{d_{ij}}\left(\mathbf{c}_i - \mathbf{c}_j\right)\Phi_x\left(\mathbf{m}_{i,j}\right) = R\mathbf{x}_i^{l+1}.
\tag{17}
$$

Therefore, we have that rotating and translating $\mathbf{c}$ results in the same rotation and no translation on $\mathbf{x}^{l+1}$ by updating through equation 7.

Thus we can conclude that the property defined in equation 15 is satisfied. $\square$

Having proved the equivariance property of a single GFN layer, then inductively, we can draw conclusion that a composition of $L$ GFN layers will also preserve the same equivariance.

## A.4 PROOF OF PROPOSITION 3

We first derive the variational lower bound (ELBO) objective in equation 8. The ELBO can be calculated as follows:

$$
\begin{aligned}
\mathbb{E}\log p_\theta(\mathcal{C}^0|\mathcal{G}) &= \mathbb{E}\log\mathbb{E}_{q(\mathcal{C}^{1:T}|\mathcal{C}^0)}\left[\frac{p_\theta(\mathcal{C}^{0:T-1}|\mathcal{G},\mathcal{C}^T)\times p(\mathcal{C}^T)}{q(\mathcal{C}^{1:T}|\mathcal{C}^0)}\right]\\
&\geq \mathbb{E}_q\log\frac{p_\theta(\mathcal{C}^{0:T-1}|\mathcal{G},\mathcal{C}^T)\times p(\mathcal{C}^T)}{q(\mathcal{C}^{1:T}|\mathcal{C}^0)}\\
&= \mathbb{E}_q\left[\log p(\mathcal{C}^T) - \sum_{t=1}^T\log\frac{p_\theta(\mathcal{C}^{t-1}|\mathcal{G},\mathcal{C}^t)}{q(\mathcal{C}^t|\mathcal{C}^{t-1})}\right]\\
&= \mathbb{E}_q\left[\log p(\mathcal{C}^T) - \log\frac{p_\theta(\mathcal{C}^0|\mathcal{G},\mathcal{C}^1)}{q(\mathcal{C}^1|\mathcal{C}^0)} - \sum_{t=2}^T\left(\log\frac{p_\theta(\mathcal{C}^{t-1}|\mathcal{G},\mathcal{C}^t)}{q(\mathcal{C}^{t-1}|\mathcal{C}^t,\mathcal{C}^0)}+\log\frac{q(\mathcal{C}^{t-1}|\mathcal{C}^0)}{q(\mathcal{C}^t|\mathcal{C}^0)}\right)\right]\\
&= \mathbb{E}_q\left[\log\frac{p(\mathcal{C}^T)}{q(\mathcal{C}^T|\mathcal{C}^0)} - \log p_\theta(\mathcal{C}^0|\mathcal{G},\mathcal{C}^1) - \sum_{t=2}^T\log\frac{p_\theta(\mathcal{C}^{t-1}|\mathcal{G},\mathcal{C}^t)}{q(\mathcal{C}^{t-1}|\mathcal{C}^t,\mathcal{C}^0)}\right]\\
&= -\mathbb{E}_q\left[\mathrm{KL}\left(q(\mathcal{C}^T|\mathcal{C}^0)\|p(\mathcal{C}^T)\right)+\sum_{t=2}^T\mathrm{KL}\left(q(\mathcal{C}^{t-1}|\mathcal{C}^t,\mathcal{C}^0)\|p_\theta(\mathcal{C}^{t-1}|\mathcal{G},\mathcal{C}^t)\right)-\log p_\theta(\mathcal{C}^0|\mathcal{G},\mathcal{C}^1)\right].
\end{aligned}
$$
(18)

It can be noted that the first term $\mathrm{KL}\left(q(\mathcal{C}^T|\mathcal{C}^0)\|p(\mathcal{C}^T)\right)$ is a constant, which can be omitted in the objective. Furthermore, for brevity, we also merge the final term $\log p_\theta(\mathcal{C}^0|\mathcal{G},\mathcal{C}^1)$ into the second term (sum over KL divergences), and finally derive that $\mathcal{L}_{\mathrm{ELBO}} = \sum_{t=1}^T D_{\mathrm{KL}}(q(\mathcal{C}^{t-1}|\mathcal{C}^t,\mathcal{C}^0)\|p_\theta(\mathcal{C}^{t-1}|\mathcal{G},\mathcal{C}^t))$ as in equation 8.

Now we consider how to compute the KL divergences as the proposition 3. Since both $q(\mathcal{C}^{t-1}|\mathcal{C}^t,\mathcal{C}^0)$ and $p_\theta(\mathcal{C}^{t-1}|\mathcal{G},\mathcal{C}^t)$ are Gaussian share the same covariance matrix $\tilde{\beta}_t I$, the KL divergence between them can be calculated by the squared $\ell_2$ distance between their means weighed by a certain weights $\frac{1}{2\tilde{\beta}_t}$. By the expression of $q(\mathcal{C}^t|\mathcal{C}^0)$, we have the reparameterization that $\mathcal{C}^t = \sqrt{\bar{\alpha}_t}\mathcal{C}^0 + \sqrt{1-\bar{\alpha}_t}\epsilon$. Then we can derive:

$$
\begin{aligned}
&\mathbb{E}_q\,\mathrm{KL}\left(q(\mathcal{C}^{t-1}|\mathcal{C}^t,\mathcal{C}^0)\|p_\theta(\mathcal{G},\mathcal{C}^{t-1}|\mathcal{C}^t)\right)\\
&= \frac{1}{2\tilde{\beta}_t}\mathbb{E}_{\mathcal{C}^0}\left\|\frac{\sqrt{\bar{\alpha}_{t-1}}\beta_t}{1-\bar{\alpha}_t}\mathcal{C}^0 + \frac{\sqrt{\alpha_t}(1-\bar{\alpha}_{t-1})}{1-\bar{\alpha}_t}\mathcal{C}^t - \frac{1}{\sqrt{\alpha_t}}\left(\mathcal{C}^t - \frac{\beta_t}{\sqrt{1-\bar{\alpha}_t}}\epsilon_\theta(\mathcal{C}^t,\mathcal{G},t)\right)\right\|^2\\
&= \frac{1}{2\tilde{\beta}_t}\mathbb{E}_{\mathcal{C}^0,\epsilon}\left\|\frac{\sqrt{\bar{\alpha}_{t-1}}\beta_t}{1-\bar{\alpha}_t}\cdot\frac{\mathcal{C}^t-\sqrt{1-\bar{\alpha}_t}\epsilon}{\sqrt{\bar{\alpha}_t}} + \frac{\sqrt{\alpha_t}(1-\bar{\alpha}_{t-1})}{1-\bar{\alpha}_t}\mathcal{C}^t - \frac{1}{\sqrt{\alpha_t}}\left(\mathcal{C}^t - \frac{\beta_t}{\sqrt{1-\bar{\alpha}_t}}\epsilon_\theta(\mathcal{C}^t,\mathcal{G},t)\right)\right\|^2\\
&= \frac{1}{2\tilde{\beta}_t}\cdot\frac{\beta_t^2}{\alpha_t(1-\bar{\alpha}_t)}\mathbb{E}_{\mathcal{C}^0,\epsilon}\left\|0\cdot\mathcal{C}^t+\epsilon-\epsilon_\theta(\mathcal{C}^t,\mathcal{G},t)\right\|^2\\
&= \frac{\beta_t^2}{2\frac{1-\bar{\alpha}_{t-1}}{1-\bar{\alpha}_t}\beta_t\alpha_t(1-\bar{\alpha}_t)}\mathbb{E}_{\mathcal{C}^0,\epsilon}\left\|\epsilon-\epsilon_\theta(\mathcal{C}^t,\mathcal{G},t)\right\|^2\\
&= \gamma_t\mathbb{E}_{\mathcal{C}^0,\epsilon}\left\|\epsilon-\epsilon_\theta(\mathcal{C}^t,t)\right\|^2,
\end{aligned}
$$
(19)

where $\gamma_t$ represent the wights $\frac{\beta_t}{2\alpha_t(1-\bar{\alpha}_{t-1})}$. And we finish the proof.

## A.5 ANALYSIS OF THE INVARIANT DENSITY IN SEC. 4.2

Given a geometric system $x\in\mathbb{R}^{N\cdot 3}$, we obtain the CoM-free $\hat{x}$ by subtracting its CoM. This can be considered as a linear transformation:

$$
\hat{x} = Qx, \quad\text{where}\quad Q = I_3\otimes\left(I_N - \frac{1}{N}\mathbf{1}_N\mathbf{1}_N^T\right) \tag{20}
$$

where $I_k$ denotes the $k\times k$ identity matrix and $\mathbf{1}_k$ denotes the $k$-dimensional vector filled with ones. It can be noted that $Q$ is a symmetric projection operator, i.e., $Q^2 = Q$ and $Q^T = Q$. And we also

have that $\text{rank}[Q] = (N-1) \cdot 3$. Furthermore, let $U$ represent the space of CoM-free systems, we can easily have that $Qy = y$ for any $y \in U$ since the CoM of $y$ is already zero.

Formally, let $n = N \cdot 3$ and set $\mathbb{R}^n$ with an isotropic normal distribution $\rho = \mathcal{N}(0, I_n)$, then the CoM-free density can be formally written as $\hat{\rho} = \mathcal{N}(0, QI_nQ^T) = \mathcal{N}(0, QQ^T)$. Thus, sampling from $\hat{\rho}$ can be trivially achieved by sampling from $\rho$ and then projecting with $Q$. And $\hat{\rho}(y)$ can be calculated by $\rho(y)$ since for any $y \in U$ we have $\|y\|_2^2 = \|Qy\|_2^2$, and thus $\rho(y) = \hat{\rho}(y)$.

And in this paper, with the SE(3)-equivariant Markov kernels of the reverse process, any CoM-free system will transit to another CoM-free system. And thus we can induce a well-defined Markov chain on the subspace spanned by $Q$.

## B    OTHER RELATED WORK

**Protein structure generation.** There has also been many recent works working on protein structure folding. For example, Boltzmann generators Noé et al. (2019) use flow-based models to generate the structure of protein main chains. AlQuraishi (2019) uses recurrent networks to model the amino acid sequences. Ingraham et al. (2019) proposed neural networks to learn an energy simulator to infer the protein structures. Most recently, AlphaFold Senior et al. (2020); Jumper et al. (2021) has significantly improved the performance of protein structure generation. Nevertheless, proteins are mainly linear backbone structures while general molecules are highly branched with various rings, making protein folding approaches unsuitable for our setting.

**Point cloud generation.** Recently, some other works (Luo & Hu, 2021; Chibane et al., 2020) has also been proposed for 3D structure generation with diffusion-based models, but focus on the point cloud problem. Unfortunately, in general, point clouds are not considered as graphs with various atom and bond information, and equivariance is also not widely considered, making these methods fundamentally different from our model.

## C    EXPERIMENT DETAILS

In this section, we introduce the details of our experiments. In practice, the means $\epsilon_\theta$ are parameterized as compositions of both typical invariant MPNNs (Schütt et al., 2017) and the proposed equivariant GFNs in Sec. 4.2. As a default setup, the MPNNs for parameterizing the means $\epsilon_\theta$ are all implemented with 4 layers, and the hidden embedding dimension is set as 128. After the MPNNs, we can obtain the informative invariant atom embeddings, which we denote as $\mathbf{h}^0$. Then the embeddings $\mathbf{h}^0$ are fed into equivariant layers and updated with equation 5, equation 6, and equation 7 to obtain the equivariant output. For the training of GEODIFF, we train the model on a single Tesla V100 GPU with a learning rate of 0.001 until convergence and Adam (Kingma & Welling, 2013) as the optimizer. The practical training time is ~48 hours. The other hyper-parameters of GEODIFF are summarized in Tab. 4, including highest variance level $\beta_T$, lowest variance level $\beta_T$, the variance schedule, number of diffusion timesteps $T$, radius threshold for determining the neighbor of atoms $\tau$, batch size, and number of training iterations.

Table 4: Additional hyperparameters of our GEODIFF.

| Task | $\beta_1$ | $\beta_T$ | $\beta$ scheduler | $T$ | $\tau$ | Batch Size | Train Iter. |
|------|-----------|-----------|-------------------|-----|--------|------------|-------------|
| QM9 | 1e-7 | 2e-3 | sigmoid | 5000 | 10Å | 64 | 1M |
| Drugs | 1e-7 | 2e-3 | sigmoid | 5000 | 10Å | 32 | 1M |

## D    ADDITIONAL EXPERIMENTS

### D.1    RESULTS FOR GEOM-QM9

The results on the GEOM-QM9 dataset are reported in Tab. 5.

Table 5: Results on the **GEOM-QM9** dataset, without FF optimization.

| Models | COV-R (%) ↑ | | MAT-R (Å) ↓ | | COV-P (%) ↑ | | MAT-P (Å) ↓ | |
|---|---|---|---|---|---|---|---|---|
| | Mean | Median | Mean | Median | Mean | Median | Mean | Median |
| CVGAE | 0.09 | 0.00 | 1.6713 | 1.6088 | - | - | - | - |
| GRAPHDG | 73.33 | 84.21 | 0.4245 | 0.3973 | 43.90 | 35.33 | 0.5809 | 0.5823 |
| CGCF | 78.05 | 82.48 | 0.4219 | 0.3900 | 36.49 | 33.57 | 0.6615 | 0.6427 |
| CONFVAE | 77.84 | 88.20 | 0.4154 | 0.3739 | 38.02 | 34.67 | 0.6215 | 0.6091 |
| GEOMOL | 71.26 | 72.00 | 0.3731 | 0.3731 | - | - | - | - |
| CONFGF | 88.49 | 94.31 | 0.2673 | 0.2685 | 46.43 | 43.41 | 0.5224 | 0.5124 |
| **GEODIFF-A** | **90.54** | **94.61** | 0.2104 | 0.2021 | 52.35 | 50.10 | 0.4539 | 0.4399 |
| **GEODIFF-C** | 90.07 | 93.39 | **0.2090** | **0.1988** | **52.79** | **50.29** | **0.4448** | **0.4267** |

Table 6: Additional results on the **GEOM-Drugs** dataset, without FF optimization.

| Models | COV-R (%) ↑ | | MAT-R (Å) ↓ | | COV-P (%) ↑ | | MAT-P (Å) ↓ | |
|---|---|---|---|---|---|---|---|---|
| | Mean | Median | Mean | Median | Mean | Median | Mean | Median |
| **GEODIFF (T=1000)** | 82.96 | 96.29 | 0.9525 | 0.9334 | 48.27 | 46.03 | 1.3205 | 1.2724 |

## D.2 ABLATION STUDY WITH FEWER DIFFUSION STEPS

We also test our method with fewer diffusion steps. Specifically, we test the setting with $T = 1000$, $\beta_1 =$1e-7 and $\beta_T =$9e-3. The results on the more challenging Drugs dataset are shown in Tab. 6. Compared with the results in Tab. 1, we can observe that when setting the diffusion steps as 1000, though slightly weaker than the performance with 5000 decoding steps, the model can already outperforms all existing baselines. Note that, the most competitive baseline CONFGF (Shi et al., 2021) also requires 5000 sampling steps, which indicates that our model can achieve better performance with fewer computational costs compared with the state-of-the-art method.

## E MORE VISUALIZATIONS

We provide more visualization of generated structures in Fig. 3. The molecules are chosen from the test split of GEOM-Drugs dataset.

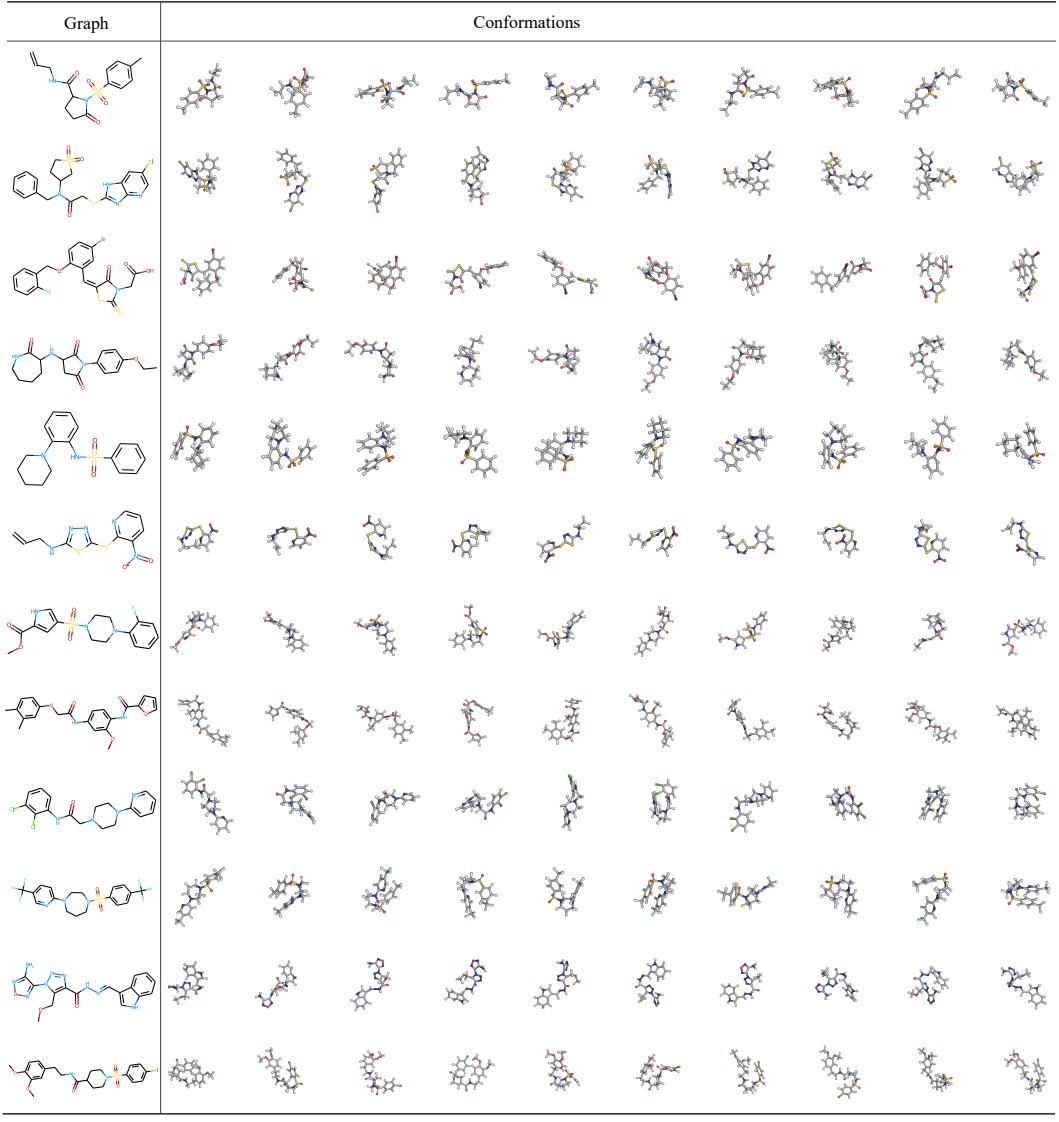

Figure 3: Visualization of drug-like conformations generated by GEODIFF.

