# OpenReview forum: "GeoDiff: A Geometric Diffusion Model for Molecular Conformation Generation"
_ICLR.cc/2022/Conference — ICLR 2022 Oral_

### Official Review · Reviewer_PLjF · 2021-10-30

**Correctness:** 4
**Technical Novelty And Significance:** 4
**Empirical Novelty And Significance:** 4
**Recommendation:** 8
**Confidence:** 3

**Main Review:**

## Pros

* GeoDiff is an end-to-end model for generating Cartesian coordinates from a molecular graph.

* GeoDiff shows very good performance on standard test sets.

* GeoDiff outperforms various state-of-the-art methods for generating molecular conformations.

## Cons

* GeoDiff's model seems to involve a large number of parameters.

## Comments and Questions

* What is the computation time for generating a structure with GeoDiff? How does this compare to other ML approaches or RDKit?

* What is the accuracy of the structures at intermediate stages of the diffusion process? Perhaps this could provide some insight into whether $T=5000$ diffusion steps are needed to reach the reported performance.

* Have you also run tests with a smaller number of diffusion steps?

* You show that GeoDiff's model and objective functions are invariant under rigid transformations. However, what if the training data are not invariant? Doesn't this still fix some arbitrary orientation in the trained model?

* Which threshold $\delta$ did you use in the computation of the precision and recall measures?

* What is the total number of model parameters that are trained?

* How is GeoDiff +FF implemented? What is GeoDiff's performance in property prediction without FF?

## Language

* Page 3: "It has been shown effective"

* Page 9: "can stay the superior diversity"

* Page 9: "yeild" instead of "yield"



**Summary Of The Paper:**

This paper describes and tests GeoDiff, an end-to-end denoising diffusion model for generating molecular conformations from a molecular graph. The idea is to learn how to denoise a random conformation so as to generate a realistic structure of a small molecule. To do so, a molecular structure is first transformed into a random configuration from an uncorrelated multivariate normal distribution via a diffusion process (forward process). By training a backwards model that inverts the diffusion process, molecular conformations can then be generated from white noise. The reverse process is modeled as a sequence of Markov kernels where each kernel is a Gaussian whose means are neural networks that are trained based on the forward  trajectories. Molecular conformations are directly parameterized in Cartesian coordinates. Therefore, an important property that should be imposed on the reverse process is *equivariance*. Equivariance refers to the following behavior of a mapping under the action of a group: Transforming the input and then mapping it to some output is the same as applying the mapping first and then transforming the output. Here, the relevant group, the special Euclidean group $SE(3)$, is formed by the rigid transformations in $\mathbb R^3$. Equivariance is implemented by drawing the initial conformation from a distribution that is invariant under rigid transformations (isotropic Gaussian that removes the center of mass) and by using equivariant Markov kernels for the reverse process. The equivariance of the Markov kernels is implemented with equivariant convolutional layers (graph field network (GFN)). Two invariant objective functions for training are derived ("alignment approach" and "chain-rule approach"). GeoDiff is tested on two standard benchmarks and shows a superior performance compared to other molecular configuration generators. Also property predictions made by GeoDiff are better than with concurrent methods.


**Summary Of The Review:**

 GeoDiff is an end-to-end model for generating Cartesian coordinates from a molecular graph. GeoDiff outperforms various state-of-the-art methods for generating molecular conformations.

---

> ### Author Response · Authors · 2021-11-19
> **Thank you for your constructive comments and suggestions! The response to your questions are listed below:**
>
> Thank you for your constructive comments and suggestions! The response to your questions are listed below:
>
> **Q1: The computational time for generation.**
> - Our generation time is in line with the most competitive existing method ConfGF (which also requires 5000 sampling steps), and requires around 8500 and 11500 seconds to decode the entire QM9 and Drugs test sets respectively. The sampling time varies among different molecules, so here we report the sampling time of the whole test set instead. This sampling time is around 100x times compared to more previous models as well as RDKit.
> - We admit that sampling inefficiency is a typical bottleneck for diffusion-based models. And as stated in the conclusion section, we leave it as an important future work about utilizing the most recent progress of diffusion methods [1] to accelerate the sampling process.
> Following this concern, we also further test an additional setting with fewer diffusion steps. Please refer to our response to your Q3 for details.
>
> **Q2: About the structures at intermediate stages.**
>
> Thanks for your suggestion and we have evaluated the intermediate structures. The results are as below:
>
> | Number of steps | COV-R(Mean) | COV-R(Median) | MAT-R(Mean) | MAT-R(Median) | COV-P(Mean) | COV-P(Median) | MAT-P(Mean) | MAT-P(Median) |
> | --- | --- | --- | --- | --- | --- | --- | --- | --- |
> |4000 | 89.33 | 97.62 | 0.8542 | 0.8251 | 61.63 | 64.33 | 1.1709 | 1.1358|
> |3000 | 71.34 | 85.39 | 1.1055 | 1.0901 | 33.57 | 25.98 | 1.5256 | 1.4822|
> |2000 | 32.40 | 22.49 | 1.3812 | 1.3756| 9.75 | 3.78| 1.8977 | 1.8495|
> |1000 | 0.65 | 0.00 | 1.9854 | 1.9965 | 0.09 | 0.00 | 2.4904 | 2.4629|
>
> By comparing with Tab.1, we can observe that running 4000 steps is already good enough and in the last 1000 steps (from 4000 to 5000) the performance only changes slightly. This interesting result indicates 4000 is enough to reach the desired performance, and we leave more detailed studies of the decoding process as future work.
>
> **Q3: Tests with fewer diffusion steps.**
>
> Thanks for your suggestion and we tested the setting with T=1000 on GEOM-Drugs. We set $\beta_1$ as 1e-7 and $\beta_T$ as 9e-3. The results are:
>
> |COV-R(Mean) | COV-R(Median) | MAT-R(Mean) | MAT-R(Median) | COV-P(Mean) | COV-P(Median) | MAT-P(Mean) | MAT-P(Median)|
> |--- |--- | --- | --- | --- | --- | --- | --- |
> |82.96 | 96.29 | 0.9525 | 0.9334 | 48.27 |  46.03 | 1.3205 | 1.2724|
>
> Compared with Tab.1, we can see T=1000 shows slightly weaker results than T=5000, but already outperforms all existing methods. Note that, the most competitive baseline ConfGF also requires 5000 sampling steps, which indicates that our model can achieve better performance with fewer computational costs compared with the state-of-the-art method.
>
> **Q4: What if the training data are not invariant?**
>
> Since our model and objective functions preserve the invariance of transformations, any arbitrary orientation will have no effect on the trained model. Therefore, even when the training data is not invariant, our model can still enjoy the invariance property.
>
> **Q5: Which threshold $\delta$ did you use?**
>
> Thanks a lot and we have missed this important information! We follow existing works (CGCF, ConfVAE and ConfGF methods mentioned in the paper) to set $\delta=0.5$ for QM9 and $\delta=1.25$ for drugs. We have added this to the paper.
>
> **Q6: The total number of model parameters.**
>
> Our current configuration holds 795858 parameters, which lies in the typical shape of graph neural networks for molecular geometric systems.
>
> **Q7: How is GeoDiff +FF implemented? What is GeoDiff's performance in property prediction without FF?**
>
> GeoDiff+FF is implemented by first drawing samples from GeoDiff as illustrated in Sec 4.4, and then further optimizing the structures with an empirical force field. Specifically, we use the API from RDKit that “from rdkit.Chem.rdForceFieldHelpers import MMFFOptimizeMolecule”.
>
> *We hope the above response and the updated draft could address your questions!*
>
> **References**: [1] Nichol, Alex, and Prafulla Dhariwal. "Improved denoising diffusion probabilistic models." arXiv preprint arXiv:2102.09672 (2021).

---

### Official Review · Reviewer_sVuD · 2021-11-02

**Correctness:** 3
**Technical Novelty And Significance:** 4
**Empirical Novelty And Significance:** Not applicable
**Recommendation:** 8
**Confidence:** 5

**Main Review:**

**Originality:** The idea of the work is novel - **GeoDiff** is the first generative model for molecular conformation generation based on a diffusion framework.

**Clarity and Quality:** The authors clearly provide motivation and state the differences of the proposed model from other approaches. The architecture is described in an easy-to-follow and intuitive way. The paper provides good **Related Work** and **Results & Discussion** sections. Still, it would be great if authors add a clear description of contributions and additional figures of architecture and an aligned cloud of conformations to visually evaluate the diversity of conformations.

**Significance:** The contribution of the paper to the field is significant. The experiments show a significant metrics gap between the proposed method and baselines, still, it would be great if authors will add standard deviation of metrics to evaluate the stability of the method.

**Drawbacks / questions**:

1. In 3.1 Authors say, they are interested in generating stable conformations. Authors provide energy metrics for the generated ensemble on GEOM-QM9 split, which covers only 30 molecules. Still, they don't provide energy metrics for medium size molecules from GEOM-DRUGS dataset.
2. Despite the proposed model being compared against a variety of baselines and that the authors mentioned the reason why they are not compared with the GeoMol, it requires a more detailed explanation of GeoMol limitations, since the GeoMol is the current state of the art approach in conformation generation task.
3. Diffusion models suffer from high computational complexity. Based on the parameters provided by the authors in appendix B, it's logical to assume that 5000 iterations per molecule are very computationally expensive. The article lacks an estimate of the generation time for one molecule.

**Summary Of The Paper:**

The work introduces a novel **GeoDiff** model for conformation generation task, based on a promising diffusion model approach that shows state-of-the-art results in other domains. The authors improved and adapted the concept of the diffusion model for the new domain so that it can work with rotation and translation invariant objects, such as conformations - authors introduced an architecture based on SE(3)-equivariant Graph Field Network (GFN) for parameterizing Markov kernels, also used roto-translational invariant starter density.

The main contributions of the paper are: the authors are the first who propose an architecture based on principal novel generative diffusion framework for molecular conformation generation and explore suitable roto-translational invariant architecture to parameterize the kernel of the method. Experiments show that the proposed model outperforms recent state-of-the-art approaches with a huge gap on **Conformation Generation** and **Property Prediction** tasks.

**Summary Of The Review:**

The idea of the paper is novel. The text of the paper is clear and easy to follow. The experiments show a significant gap in metrics values between the proposed model and baselines. Still, there are several minor drawbacks that can be fixed in the next revision of the paper.

---

> ### Author Response · Authors · 2021-11-19
> **Thank you for your constructive comments and suggestions! The response to your questions are listed below:**
>
> Thank you for your constructive comments and suggestions! The response to your questions are listed below:
>
> **Q1: About generating stable conformations.**
>
> - First of all, one could note that the GEOM dataset provides stable conformations with local minimum energy for all molecules. Therefore, by saying “generating stable conformations”, we mean our model can fit the distribution of the dataset and generate conformations close to the dataset, which is verified by experiments in Sec 5.2.
> - Besides, one should notice that the energy prediction task is not used to test “whether generated conformations are stable with low enough”, but to verify “whether the energy ensemble of generated ones is near to reference ones”. Therefore, the energy metrics are not directly related to the structure stability.
> - **For the dataset setup of energy prediction task**: We just directly follow previous work ConfGF to compare over 30 molecules of GEOM-QM9, which is a standard benchmark. We have also followed your suggestion to test on GEOM-Drugs, but the results turned out to be meaningless with extremely high variance. We think this is because small changes of large molecules can lead to a significant difference of energy ensemble, and thus not suitable for this benchmark.
>
> **Q2: Detailed explanation of GeoMol limitation.**
>
> GeoMol requires that the molecular graph should contain a special substructure of the dihedral pattern. Specifically, in the official implementation, the pattern is set with open-sourced RDKit tool as “dihedral_pattern = Chem.MolFromSmarts('[\*]\~[\*]\~[\*]\~[\*]')”, which filtered out about one-third of the dataset and made it incomparable with other models.
>
> **Q3: An estimation of the generation time.**
>
> - Our generation time is in line with the most competitive existing method ConfGF (which also requires 5000 sampling steps), and requires around 8500 and 11500 seconds to decode the entire QM9 and Drugs test sets respectively. The sampling time varies among different molecules, so here we report the sampling time of the whole test set instead.
> - We admit that sampling inefficiency is a typical bottleneck for diffusion-based models. And as stated in the conclusion section, we leave it as an important future work about utilizing the most recent progress of diffusion methods [1] to accelerate the sampling process.
> - Following your concern, we also test an additional setting with T=1000 on GEOM-Drugs. We set $\beta_1$ as 1e-7 and $\beta_T$ as 9e-3. The results are:
> |COV-R(Mean) | COV-R(Median) | MAT-R(Mean) | MAT-R(Median) | COV-P(Mean) | COV-P(Median) | MAT-P(Mean) | MAT-P(Median)|
> | --- | --- | --- | --- | --- | --- | --- | --- |
> | 82.96 | 96.29 | 0.9525 | 0.9334 | 48.27 |  46.03 | 1.3205 | 1.2724 |
> - This can further speed up GeoDiff by 5 times. Compared with Tab.1, we can see T=1000 shows slightly weaker results than T=5000, but already outperforms all existing methods. Note that, the most competitive baseline ConfGF also requires 5000 sampling steps, which indicates that our model can achieve better performance with fewer computational costs compared with the state-of-the-art method.
>
> We hope the above response and the updated draft could address your questions!
>
> **Reference**: [1] Nichol, Alex, and Prafulla Dhariwal. "Improved denoising diffusion probabilistic models." arXiv preprint arXiv:2102.09672 (2021).

---

> > ### Comment · Reviewer_sVuD · 2021-11-25
> > **Thank you for answering questions and providing additional experiments.**
> >
> > The authors clarified all unclear parts, accelerated the sampling process, and provided an estimate of the sampling time. I will raise my score from 6 to 8.

---

### Official Review · Reviewer_s7Z3 · 2021-11-03

**Correctness:** 3
**Technical Novelty And Significance:** 3
**Empirical Novelty And Significance:** 3
**Recommendation:** 6
**Confidence:** 3

**Main Review:**

This paper brings together recent ideas and methods (e.g. diffusion, SE(3) equivariance) to the established task of molecular conformation generation with impressive empirical results. A few more experiments are necessary. The paper is well written. I am only tangentially in this area so I am not intimately familiar with the prior works and benchmarks, and may have missed something in the empirical evaluation.

Comments:
* The authors state that roto-translation invariance is a critical inductive bias for this problem. But I am not convinced that it is critical for this setup and model class. It would be nice to include a baseline of the diffusion model without SE(3) invariance.
* I am curious about the sensitivity of the performance of the model to the hyperparameters of the diffusion process. Can the authors describe how they reached the current settings given in Table 5. In particular, T=5000 steps seems quite high and I was curious about the performance with lower T say around 1000. As a minor note, there appears to be an inconsistency in the reported batch size (128 in the text vs. 32/64 in Table 5)
* What is the generation time with 5000 diffusion steps? Does that become a bottleneck in conformation generation?
* What happened to the other methods on the geom-drugs dataset for Table 3?
* The authors should mention recent point cloud diffusion papers in their related works as it is addressing a very similar problem
* What are the reverse process sigmas set to?

Minor comments:
* Typo: translatoinal
* Typo: sophisticate
* I noticed a reference that seemed incorrect: Xu et al 2021a as the reference for molecular generation with VAEs in the second paragraph. The authors should double check their references


**Summary Of The Paper:**

This paper tackles the problem of conditional generation of molecular conformations (i.e. 3D cartesian atom positions) given a molecular graph. The authors formulate the generation process via diffusion probabilistic models; Conformations are generated by learning a reverse diffusion process from isotropic gaussian noise to molecular conformations. They use a SE(3) invariant formulation of the diffusion process based on Kohler et al 2020, and they operate directly on atomic positions (i.e. a point cloud) instead of interatomic distances or an intermediate bond geometry representation. The authors show state of the art results evaluated by COV/MAT metrics on GEOM-Drugs and GEOM-QM9 datasets.

**Summary Of The Review:**

Overall a nice paper with technical novelty and good empirical results. I have some questions about the evaluation since some apt baselines are missing. I will raise my score if the authors can address my comments.

---

> ### Author Response · Authors · 2021-11-19
> **Thank you for your constructive comments and suggestions! The response to your comments are listed below:**
>
> Thank you for your constructive comments and suggestions! The response to your comments are listed below:
>
> **Q1: Not convinced about the equivariant setup and it would be nice to include a baseline without SE(3) invariance.**
>
> Thanks for your suggestion, and we have designed an additional non-SE(3) equivariant baseline. We use the same networks architecture to obtain atom embeddings $h$ (as in Eq.6), and then directly output the atomic $\epsilon$ from $h$ by MLPs. Therefore, $\epsilon$ is non-equivariant w.r.t $C$. The numerical results on the challenging GEOM-Drugs dataset are as below:
>
> |COV-R(Mean) | COV-R(Median) | MAT-R(Mean) | MAT-R(Median) | COV-P(Mean) | COV-P(Median) | MAT-P(Mean) | MAT-P(Median)|
> | --- | --- | --- | --- | --- | --- | --- | --- |
> |0.0 | 0.0 | 54.8588 | 55.0188 | 0.0 | 0.0 | 634.4928 | 625.7850 |
>
> As expected, the non-equivariant model shows extremely poor performance, which is in line with the observation in previous works [1].
>
> **Q2: How to select the hyper-parameters. How about lower diffusion steps T like 1000.**
>
> - The majority of hyper-parameters ($T$, $\tau$, batch size, and training iters) are selected by following the previous work ConfGF. For the beta scheduler, our strategy is to ensure the variances from 0 to T go exponentially from around 0.01 to 10, which has also shown effectiveness in ConfGF.
> - The most competitive baseline ConfGF also uses 5000 sampling steps, where they set 50 noise scales and conduct 100 optimization steps for each noise scale. Therefore we also choose T=5000. We have followed your suggestion to test the setting with T=1000 on GEOM-Drugs. We set $\beta_1$ as 1e-7 and $\beta_T$ as 9e-3. The results are:
> |COV-R(Mean) | COV-R(Median) | MAT-R(Mean) | MAT-R(Median) | COV-P(Mean) | COV-P(Median) | MAT-P(Mean) | MAT-P(Median)|
> | --- | --- | --- | --- | --- | --- | --- | --- |
> | 82.96 | 96.29 | 0.9525 | 0.9334 | 48.27 |  46.03 | 1.3205 | 1.2724 |
> - Compared with Tab.1, we can see when T=1000, though slightly weaker than T=5000, it already outperforms all existing methods. Note that, the most competitive baseline ConfGF also requires 5000 sampling steps, which indicates that our model can achieve better performance with fewer computational costs compared with the state-of-the-art method.
>
> **Q3: About the generation time.**
>
> Our generation time is in line with the most competitive existing method ConfGF, and it requires around 8500 and 11500 seconds to decode QM9 and Drugs test sets respectively. This is a typical bottleneck for diffusion-based models. And as stated in the conclusion section, we leave it as an important future work about utilizing the most recent progress of diffusion methods [2] to accelerate the sampling process.
>
> **Q4: About other methods on Tab.4.**
>
> This table mainly aims to compare GeoDiff with RDKit, and thus we omit other models. We have additionally tested the performance of several other methods. The results are as follows:
>
> |Method | COV-R(Mean) | COV-R(Median) | MAT-R(Mean) | MAT-R(Median) | COV-P(Mean) | COV-P(Median) | MAT-P(Mean) | MAT-P(Median)|
> | --- | --- | --- | --- | --- | --- | --- | --- | --- |
> |GraphDG+FF | 90.71 | 100.00 | 0.8839 | 0.8514 | 57.21 | 60.81 | 1.4549 | 1.4268|
> |CGCF+FF | 90.56 | 100.00 | 0.7827 | 0.7573 | 74.06 | 81.88 | 1.1736 | 1.1230|
> |ConfVAE+FF | 91.43 | 100.00 | 0.7749 | 0.7575 | 75.74 | 83.32 | 1.1391 | 1.1007|
> |ConfGF+FF | 92.54 | 100.00 | 0.7696 | 0.7528 | 76.34 | 85.86 | 1.1374 | 1.0886|
>
> Compared with the results in Tab.3,  we can see that when further optimized with FF, some baselines can achieve similar performance considering the Recall (-R) metrics, but still behind GeoDiff with a large margin for the Precision (-P) metrics.
>
> **Q5: Mention related work for the point cloud generation.**
>
> Thanks for the suggestion. We agree that the problem is related and have added them now. We would highlight that for existing point cloud generative models,  typically the data is not in graph structures with atom and bond information, and the equivariance is not considered, making them fundamentally different from our method.
>
> **Q6: What are sigmas?**
>
> In this work we set $\sigma_t = 1 - \frac{\alpha_t}{\alpha_{t-1}}$.
>
> ------------------------------------------------------------
>
> We hope the above response and the updated draft could address your questions!
>
> **References:**
>
> - [1] Mansimov, Elman, et al. "Molecular geometry prediction using a deep generative graph neural network." Scientific reports 9.1 (2019): 1-13.
> - [2] Nichol, Alex, and Prafulla Dhariwal. "Improved denoising diffusion probabilistic models." arXiv preprint arXiv:2102.09672 (2021).

---

### Public Comment · ~Yiyi_Joe1 · 2021-11-16
**Some questions about the theoretical analysis provided in this paper**

Thanks for the submitted work. Here are some questions about the key theoretical analysis provided in this paper. We think these following questions are necessary for authors to respond.

Question 1:

In page 5, a claim states that “$\hat{\rho}(C)$ is translation-invariant”. However, for $x \in U$, if $\hat{\rho}(x+g) = 0$ while $\hat{\rho}(x) \not= 0$, how could $\hat{\rho}(C)$ be translation invariant? Looking forward to authors' response.

Question 2:

In page 5, for showing Markov Kernels equivariant, $\epsilon_\theta$ is clearly roto-translation equivariant, however, $\mu_{\theta}(RC+g, t) = R\mu_{\theta}(C, t) + g/\sqrt{\alpha}$ is not equivariant. So is Markov Kernel.

$\mu_{\theta}(RC^{t}+g, t) = \frac{1}{\sqrt{\alpha_{t}}}(RC^{t}+g - \frac{\beta_{t}}{\sqrt{1-\overline{\alpha_{t}}}}\epsilon_{\theta}(\mathcal{G}, RC^{t}+g, t))$

$= \frac{1}{\sqrt{\alpha_{t}}}(RC^{t}+g - \frac{\beta_{t}}{\sqrt{1-\overline{\alpha_{t}}}}(R\epsilon_{\theta}(\mathcal{G}, C^{t}, t)))$

$= \frac{1}{\sqrt{\alpha_{t}}}R(C^{t}+\frac{g}{R} - \frac{\beta_{t}}{\sqrt{1-\overline{\alpha_{t}}}}(\epsilon_{\theta}(\mathcal{G}, C^{t}, t))))$

$= \frac{1}{\sqrt{\alpha_{t}}}R(C^{t} - (\frac{\beta_{t}}{\sqrt{1-\overline{\alpha_{t}}}}\epsilon_{\theta}(\mathcal{G}, C^{t}, t))) + \frac{1}{\sqrt{\alpha_{t}}}g$

$\not= \frac{1}{\sqrt{\alpha_{t}}}R(C^{t} - (\frac{\beta_{t}}{\sqrt{1-\overline{\alpha_{t}}}}\epsilon_{\theta}(\mathcal{G}, C^{t}, t))) + g$

$= R\mu_{\theta}(C^{t}, t) + g$

How could $\mu_{\theta}$ be roto-translation equivariant? The only valid case is that $C^{t}$ is translation-invariant. However, $C^{t}$ is just coordinate. Could authors provide a formal proof of this claim?

Question 3:

In page 6, section "Alignment approach", you mention that "conformation $C^{0}$ is equivariant with $C^{t}$, the processed $\hat{\epsilon}$ will also enjoy equivariance". This claim is not clear. When you mention equivariance, please also provide the function and the variable?

Question 4:

In page 7, section "Chain rule approach", you mention that "our practical implementation is to first approximately calculate $\nabla_{d^{t}}q(C^{t}|C^{0})$ as $\frac{d^{t}-\sqrt{\overline{\alpha_{t}}}d^{0}}{1-\overline{\alpha_{t}}}$". What is the exact gradient? Is the approximation error small enough? Looking forward to further clarifications.

Question 5:

Based on equation (20) in page 15, Q is in $R^{3N\times 3N}$, x is in $R^{N\times 3}$. So how can $Qy = y$? The equation (20) is clearly incorrect.

Question 6:

In page 16, an isotropic normal distribution $\rho = N(0, I_{3})$. Why $\rho$ is defined on $R^{3}$, not $R^{3N}$? Since a sample should be a conformer, not a single coordinate? You also mention that $\hat{\rho}(y) = \rho(y)$, why is this case? In this case, $QQ^{T} = I_{3}$, but clearly this is not correct. So $\hat{\rho}(y) = c\rho(y)$ for some constant $c$? Quite confused about these claims. Looking forward to authors' response.

We are really expecting the authors to provide formal proof and explanations to the above questions. Sincerely thanks!

---

> ### Author Response · Authors · 2021-11-19
> **Thank you very much for your comments! The response to your questions are listed below:**
>
> Thank you very much for your comments! The response to your questions are listed below:
>
> **Q1: How $\hat{\rho}(C)$ ensures translation invariance?**
>
> - Note that, in Sec 4.2 we stated that we consider the CoM-free system. As illustrated in the “invariant density” paragraph, we will first move $C$ to zero CoM for calculating the likelihood. Therefore, structures with different translations will always be of the same probability.
> - Then, let’s consider your example. First you assume $x \in U$. As in Appendix A.5, $U$ is defined as zero-CoM subspace. Therefore, if you have $\hat{\rho}(x+g)=0$, then $x+g$ must be a zero-CoM system, and therefore $\hat{\rho}(x)=\hat{\rho}(x+g)=0$.
>
> **Q2: How Markov kernels ensure translation equivariance?**
>
> - Again, we would highlight that as stated in the paper, we also always consider the CoM-free system in equivariant kernels. Therefore, just as your speculation that “the only valid case is that $C_t$ is translation-invariant”, $C_t$ indeed is translationally invariant since we always first move them to zero-CoM, and therefore we hold the equivariance.
> - Furthermore, in practice, this condition is also very easy to implement. As long as the initial structure $C_T$ is sampled as a zero-CoM system, along the way all intermediate structures $C_t$ will still enjoy a zero-CoM with the equivariant updates and therefore naturally meet the above requirement.
>
> **Q3: About alignment approach.**
>
> Yes, here actually we implicitly define the function and variable. You can view this as a function $f(C_t; C_0)$ that outputs a new coordinate as $\hat{C}_0$ that is equivariant with the input $C_t$.
>
> **Q4: About chain-rule approach.**
>
> - As shown in ConfGF, when computing $\nabla_{d^t} p(C^t|C^0)$ they just compute it as $\nabla_{d^t} p(d^t|C^0)$. Here, in our formulation, \$p(d^t|C^0)$ are intractable distributions so we cannot calculate the exact gradient. We just follow the convention of machine learning to take this distribution as Gaussians and compute the $L^2$ distance as gradients.
> - For the question of whether the Gaussian assumption is a good approximation, I think this is a more foundational problem of ML. This is more related to the empirical results, and for this work in practice this approximation works pretty well and leads to competitive performance.
>
> **Q5: About shapes of matrices in Eq.20.**
>
> Thank you very much for your check and this is a typo! $x$ should be in $N \cdot 3$ instead of $N \times 3$. We have updated it in the paper.
>
> **Q6: About the theoretical analysis of the invariant Gaussian.**
>
> *Q6.1: Why $\rho$ is defined on $R^3$ instead of $R^{3N}$?*
>
> - Thank you very much and there are some typos here. $\rho$ should lie in $R^{3N}$. We have updated the paper and you can check it now.
>
> *Q6.2: Why $\hat{\rho}(x)=\rho(x)$?*
>
> - As illustrated in the paper, we hold this equation under the condition that $x \in U$. This is used to show that we can just calculate the likelihood of a CoM-free system by normal Gaussian. More specifically, since $||x||^2_2 = ||Qx||^2_2$, we have that ${\rho}(x) = \hat\rho(Qx) = \hat\rho(x)$. So this equation didn’t means that $\hat{\rho}(x)$ and $\rho(x)$ are two same distribution.
>
> *We hope the above response and the updated draft could address your questions!*

---

### Author Response · Authors · 2021-11-19
**General response to all reviewers and public readers**

We would like to thank all the reviewers and public readers for your constructive feedback. We’ve updated the paper according to your suggestions. More specifically, we have made the following changes:

- We add several additional contents to the paper, including extra experiments and related works. We have also added several additional experiment details. These contents are all marked in blue color. Due to the page limits, most contents are added to the Appendix part.
- We have updated several typos.

*We hope the above response and the updated draft could address your questions!*

---

### Public Comment · ~Octavian-Eugen_Ganea1 · 2022-02-01
**Regarding excluding the SOTA baseline GeoMol (Ganea et al, NeurIPS 2021)**

First of all, congratulations for the acceptance of your great work!

We would like to reply to your decision of not including the SOTA baseline GeoMol (Ganea et al, NeurIPS 2021) among your baselines. We noted your reason stated below that GeoMol uses the dihedral pattern '[ * ] ~ [ * ] ~ [ * ] ~ [ * ]') which you claim to "filter out about one-third of the dataset".  We show that the percentage of errors from GeoMol is, in fact, below 0.7%. First, the above pattern is simply saying that the molecule must have at least 4 atoms, 4 bonds and have one rotatable bond, otherwise the molecule is almost trivial to predict even without any ML model. Second, we added the following jupyter notebook [1] to our repo to count the actual number of SMILES that cannot be processed by GeoMol from both GEOM-QM9 and GEOM-DRUGS. As you can see, GeoMol cannot generate conformes for only 62 out of 10K random QM9 SMILES (sampled randomly from the full 133258 sized dataset), and for only 1 out of 10K random DRUGS SMILES (randomly from the full 304466 sized dataset).

We kindly ask you to correct your main paper's text regarding the reason of not using GeoMol as a valid baseline, or, to add GeoMol as a baseline. We hope that future research papers will use both GeoDiff and GeoMol as strong baselines to improve upon! Thanks for your understanding.

[1] https://github.com/PattanaikL/GeoMol/blob/main/count_geomol_failures.ipynb

---

> ### Public Comment · ~Minkai_Xu1 · 2022-02-01
> **Thank you for your kind comments!**
>
> Hi Octavian-Eugen,
>
> Thanks for your comments here! We will follow your suggestion and rewrite the part about the concurrent work GeoMol in the final version!! The Jupyter notebook clearly shows the data was not filtered much by the dihedral pattern.
>
> In our experiments, we just observed a fraction of filtering in the data-loader and thus make the statement. You clearly show that this is not because of the pattern matching, and we will re-study this observation carefully.
>
> GeoMol indeed presents a great contribution to this community. Thanks again for your kind comment!

---

### Public Comment · ~Yangliao_Geng1 · 2022-04-01
**Regarding the proof of Proposition 1**

Thank you for nice work. There seems to be an integration-by-substitution step in the proof of Proposition 1.

In the first line of Equ. (14), some new random variables (T_g(x_0), T_g(x_1), ..., T_g(x_T)) are introduced.  Correspondingly, there should be a volume change |det (T_g)^{-1}| multiplying each infinitesimal d x_i in the integration. But in this work, T_g is a rotation matrix, so |det (T_g)^{-1}| happens to be 1. Therefore, although the above-mentioned volume change factor is omitted in the proof, the coclusion still holds.

Is my understanding correct?

---

> ### Public Comment · ~Minkai_Xu1 · 2022-04-01
> **Yes you are right**
>
> Yes, you are right. Since every $T_g$ can be represented by a SO(3) matrix and thus det($T_g$) is well-defined. In our work, | det ($T_g^{-1}$) | is manifest to be 1 so this is omitted in our proof.
>
> More generally, let's consider the case $T_g$ be a general representation of any group, actually we always have the similar conclusion. Let p(x) still be an invariant density:
>
> $$
> 1 =\int p(x) d x
> =\int \rho(T_g(y))|\operatorname{det} T_g| d y
> =\int \rho(y)|\operatorname{det} T_g| d y
> =|\operatorname{det} T_g| \underbrace{\int \rho(y) d y}_{=1}
> =|\operatorname{det} T_g|
> $$
>
> Hope this could address your questions!

---

> > ### Public Comment · ~Yangliao_Geng1 · 2022-04-01
> > **Got it. Thanks.**
> >
> > Got it. Thanks.

---

### Decision · Program_Chairs · 2022-01-20

**Decision:**

Accept (Oral)

**Comment:**

The authors focus on the conditional generation of molecular conformations (i.e. 3D cartesian atom positions) from a given molecular graph. They formulate the generation via diffusion probabilistic models.  Conformations are generated by a reverse diffusion process from isotropic Gaussian noise to molecular conformations. This diffusion process is learned from data using a SE(3) invariant formulation of the diffusion process. The authors work directly with atomic positions (i.e. a point cloud) instead of interatomic distances or an intermediate bond geometry representation. Experimental evaluations show state-of-the-art results according to COV/MAT metrics on GEOM-Drugs and GEOM-QM9 datasets.

Strengths:

- High technical novelty: first generative model for molecular conformation generation based on a diffusion framework
- Very clearly written paper.
- Impressive empirical results with state-of-the-art results on GEOM-Drugs and GEOM-QM9 datasets.

Weaknesses:

- Most of the weaknesses reported by the reviewers seem to have been addressed in the rebuttal.

The idea of the work is highly novel. The authors propose the first generative model for molecular conformation generation based on a diffusion framework. This paper brings together recent ideas and methods (e.g. diffusion, SE(3) equivariance) to the established task of molecular conformation generation with impressive empirical results. All the reviewers agree on acceptance with high scores. I recommend the authors to look at the reviewers' comments to improve the paper for the camera-ready version